Review Article

# The DNA dialect: a comprehensive guide to pretrained genomic language models

Marcell Veiner [ID][1] & Fran Supek [ID][1,2][✉]

## Abstract

Following their success in natural language processing and protein biology, pretrained large language models have started appearing in genomics in large numbers. These genomic language models (gLMs), trained on diverse DNA and RNA sequences, promise improved performance on a variety of downstream prediction and understanding tasks. In this review, we trace the rapid evolution of gLMs, analyze current trends, and offer an overview of their application in genomic research. We investigate each gLM component in detail, from training data curation to the architecture, and highlight the present trends of increasing model complexity. We review major benchmarking efforts, suggesting that no single model dominates, and that task-specific design and pretraining data often outweigh general model scale or architecture. In addition, we discuss requirements for making gLMs practically useful for genomic research. While several applications, ranging from genome annotation to DNA sequence generation, showcase the potential of gLMs, their use highlights gaps and pitfalls that remain unresolved. This guide aims to equip researchers with a grounded understanding of gLM capabilities, limitations, and best practices for their effective use in genomics.

**Keywords** Genomic Language Models; Deep Learning in Genomics; DNA and RNA Sequence Modeling; Variant Effect Prediction; Self-Supervised Learning
**Subject Categories** Chromatin, Transcription & Genomics; Computational Biology

## Introduction

In recent years, a number of advancements (Khurana et al, 2023) have propelled the field of natural language processing (NLP) to the forefront of science. Large language model (LLM) frameworks, such as the generative pretrained transformer (GPT) (preprint: Radford et al, 2018) or bidirectional encoder representations from transformers (BERT) (Devlin et al, 2019), have led to human-level performance on several NLP related tasks, such as speech recognition (Gulati et al, 2020), translation (Vaswani et al, 2017), or

question answering (preprint: Achiam et al, 2024). These models leverage millions (or even billions) of parameters to produce accurate responses, which in turn, require massive amounts of data for training. Self-supervised learning (Liu et al, 2021) is an attempt to mitigate this issue, whereby models are first pretrained on unlabeled data, enabling them to learn generally valid syntax and semantics, before being applied to specific downstream tasks. For this reason, models that underwent extensive pretraining are often termed foundation models (FMs) (preprint: Bommasani et al, 2022), which can then be readily adapted to domain-specific applications through transfer learning scenarios.

In the biological domain, the LLM paradigm was also successfully applied to protein sequences (see Guo et al, 2025; preprint: Liu et al, 2025a; Sarumi and Heider, 2024). Models such as the ESM family (Lin et al, 2023), ProteinBERT (Brandes et al, 2022), or ProtTrans (Elnaggar et al, 2022) learn content-aware data representations from large-scale protein databases (e.g., UniProt (The UniProt Consortium et al, 2025)), shaped by evolutionary pressures that manifest in the distribution of amino acid sequences in natural proteins. Consequently, patterns learned by protein language models (pLMs) are useful in designing new proteins with desirable properties (Ferruz et al, 2022), predicting chemical properties from sequence (Brandes et al, 2022), or inferring 3D structure (Lin et al, 2023).

Following the successes of pLMs, genomic language models (gLMs) (also called genomic foundation models), trained on DNA or RNA, were put forth, aiming to enhance genomic prediction and design tasks (Benegas et al, 2025c). One of the earliest gLMs was DNABERT (Ji et al, 2021), trained on the human reference genome, which showed promising capabilities on predicting transcription factor binding sites, among others. Similarly, RNABERT (Akiyama and Sakakibara, 2022) was the first gLM trained on RNA sequences, designed for structural alignment and clustering tasks. Bridging the gap between the languages of nucleic acids and amino acids, models like GenSLMs (Zvyagin et al, 2022) and cdsBERT (preprint: Hallee et al, 2023) operate on the level of codons, highlighting growing interest in codon-level representations as a complementary approach to modeling biological sequences (Boshar et al, 2024).

Since then, a large number of gLMs have appeared with different architectures (Benegas et al, 2025c), trained on the genomes of various organisms (Nguyen et al, 2024) or even individuals of a population (Dalla-Torre et al, 2025). Each approach has made different design choices, including the type of training data, the amount of information it can process at one time, and its

[1]Institute for Research in Biomedicine (IRB Barcelona), Barcelona, Spain. [2]Biotech Research & Innovation Centre (BRIC), University of Copenhagen, Copenhagen, Denmark.
[✉]E-mail: fran.supek@bric.ku.dk

**Glossary**

**Ablation (study)**
A method for evaluating the importance of specific components in a machine learning model by removing or altering them and measuring the effect on performance.

**Attention**
A mechanism that allows the model to weigh and focus on different parts of a sequence when making predictions.

**BERT**
Bidirectional Encoder Representations from Transformers; a bidirectional transformer model pretrained with masked language modeling, used for language understanding tasks like classification and question answering.

**CLM**
Causal Language Modeling; a training task where the model predicts the next token in a sequence based only on previous tokens (unidirectional).

**CNN**
Convolutional neural network; a model architecture that detects local patterns in sequences using sliding filters, which can learn to recognize biologically meaningful motifs, analogous to position weight matrices, in genomic data.

**Contrastive Learning**
A training approach that learns to distinguish similar from dissimilar pairs by pulling representations of related inputs closer and pushing unrelated ones apart.

**Decoder**
The component that generates outputs from encoded representations, often used in generative tasks.

**Embedding**
A numerical representation of a token (e.g., nucleotide or k-mer) in a continuous vector space.

**Encoder**
The component of a model that processes input sequences into internal representations.

**Few-shot Prediction**
Adapting a model to a new task using a small number of labeled examples in the input sequence.

**Fine-tuning**
Adapting a pretrained model to a specific task using labeled data, by changing its weights via further training.

**gLM**
A genomic language model pretrained in a self-supervised manner on DNA or RNA sequences to learn generalizable representations of genomic patterns, which can be transferred to downstream biological tasks.

**GPT**
Generative pretrained Transformer; a unidirectional transformer model pretrained to predict the next token, used for generative tasks such as text completion and summarization.

**LLM**
Large language model; a deep learning model trained on vast amounts of text data to understand and generate human-like language.

**Mixed Modal Approach**
A modeling strategy where different data modalities (DNA, RNA, Protein) are combined into a shared representation space, allowing the model to process them jointly.

**MLM**
Masked Language Modeling; a self-supervised training task where some tokens in a sequence are masked and the model learns to predict them.

**Multimodal Approach**
Unlike Mixed Modal, Multimodal approaches operate on the already processed representations of different modalities (often by domain-specific encoders) to combine them.

**NLP**
Natural language processing; a field of artificial intelligence focused on enabling computers to understand, interpret, and generate human language.

**pLM**
A protein language model trained on amino acid sequences to learn representations of protein structure or function, analogous to genomic language models.

**Pretraining**
Training a model on a large corpus of unlabeled genomic sequences to learn general representations, for example by self-supervised learning.

**Probing**
Evaluating what information is captured in a model's embeddings or layers using simple downstream classifiers.

**RNN**
Recurrent neural network; processes sequences sequentially, maintaining a memory of previous inputs. Variants such as LSTM (Long Short-Term Memory) and GRU (Gated Recurrent Unit) are designed to better capture long-range dependencies.

**Self-supervised Learning**
Learning from unlabeled data by creating surrogate tasks that guide representation learning, e.g. MLM or CLM.

**SSM**
State-space model; model for sequential data using linear dynamical systems, offering efficient alternatives to traditional sequence models.

**Token**
A fixed-length unit of a (DNA or RNA) sequence (e.g., nucleotide or k-mer) used as input to LLMs.

**Transfer Learning**
The process of adapting a pretrained model to a new, related task by fine-tuning it on task-specific data, enabling faster learning and better performance.

**Transformer**
A deep learning architecture based on attention mechanisms, effective for modeling pairwise dependencies in sequences.

**Zero-shot Prediction**
Applying a model to a task without any additional training on task-specific data, or adaptation of its output.

computational needs, which all have an impact on the applicability and performance of the model. In this review, we discuss the most important components of gLMs, including the genomic data used during training, the tokenizing strategy, pretraining task, architecture, and parameter counts. Next, we discuss the effective use of gLMs for genomic research, with strategies such as zero-shot prediction, in-context learning, and fine-tuning, highlighting their flexibility and effectiveness in diverse genomic applications. We also cover the importance of benchmark datasets for evaluating model performance, as well as ongoing work in improving model performance, biases, and the computational costs of training and using large models. Finally, we explore the diverse applications of gLMs in research, that is, their role in biological sequence generation, genomic sequence annotation, gene expression regulation modeling, variant effect prediction, RNA structure prediction, and metagenomics tasks.

# Taxonomy of genomic language models

In this section, we examine the pivotal design choices behind gLMs, including pretraining data, tokenizers, and architecture, and how they influence downstream performance. We focus our analysis to self-supervised models trained on DNA or RNA sequences, explicitly excluding pLMs. While large sequence-to-function models like Enformer (Avsec et al, 2021), Borzoi (Linder et al, 2025), and BigRNA (preprint: Celaj et al, 2023) offer potential for domain-specific adaptation, they rely on supervised learning. This

distinction also applies to the recently released AlphaGenome (preprint: Avsec et al, 2025). AlphaGenome unifies multimodal prediction to resolve diverse molecular phenotypes, such as splicing and chromatin states, at base-pair resolution from 1 Mb contexts. Unlike gLMs, which learn generalizable representations via self-supervision on DNA, these sequence-to-function models are trained in a supervised manner directly to predict experimental functional tracks. Therefore, these architectures are not covered in this review.

In many cases, gLMs represent a family of models, all released in the same publication, with various configurations having different parameter counts, and/or pretrained on different datasets. In general, we will refer to the model family, highlighting the specific configuration when needed. A comprehensive list of published models and their design choices is available in Dataset EV1. The data modalities and common applications of selected gLMs are presented in Fig. 1 and Table 1, while their architectural choices are presented in Table 2. Commonalities and differences between genomic and natural language modeling are highlighted in Box 1.

## Data modality

### Whole-genome approaches

GLMs require large amounts of data to be pretrained. For this reason, most gLMs were trained on the whole-genome sequences of different species, for example, on the human reference genome, e.g., DNABERT (Ji et al, 2021), and GROVER (Sanabria et al, 2024a). The context size or receptive field of these models, the length of the input sequence that the model can effectively consider when making predictions, varies from a few hundred base pairs up to 1 Mb (see Dataset EV1). Thus, context size is one of the most important factors in determining the applicability of gLMs, along with the type and amount of data a gLM has been pretrained on.

Some results indicate that training on genomes of multiple species is beneficial for improving model performance and generalization (Dalla-Torre et al, 2025; de Almeida et al, 2025a, preprint: Wu et al, 2025, Tomaz da Silva et al, 2025). It is likely that larger models benefit from this training more, as they have a greater capacity to represent cross-species diversity and variation (preprint: Wu et al, 2025; Zhai et al, 2025). Consequently, the largest gLMs to date utilize genomes from across the tree of life: CD-GPT (preprint: Zhu et al, 2024), DNAGPT (Zhang et al, 2023), Evo (Nguyen et al, 2024). On the other hand, domain-specific models are also available, e.g., for plant genomes (see also Xu et al, 2025): PlantCaduceus (Zhai et al, 2025); for fungi: Species-LM (Karollus et al, 2024); for bacteria/phages: MegaDNA (Shao and Yan, 2024), and more (Dataset EV1).

### Incorporating genetic variation

Another source of data can be obtained from focusing on genetic variation between individuals of the same species or between related species. To incentivize the model to learn patterns of genetic variation, GPN-MSA (Benegas et al, 2025a) relies on constructing a multiple sequence alignment (MSA) matrix across genomes of diverse species, thus highlighting evolutionary constraints and genetic polymorphisms. Nonetheless, the construction of MSA can be computationally costly, and is not always feasible based on the desired input length, species, and most importantly the genomic region of interest (with paralogs and repetitive DNA,

---

**Box 1    Genomic language modeling (GLM) vs. natural language processing (NLP)**

<u>Nature of data</u>
   **(NLP):** *Human language in written form, from texts like books, articles or conversations, often collected from the internet in bulk.* **(GLM):** *Sequences of DNA, RNA, or proteins encoding biological information. Whole-genome sequences of single organisms, or collections of sequences from a population or different species.*

<u>Patterns of interest</u>
   **(NLP):** *Focuses on syntactic structures, word relationships, and dependencies within text, capturing meaning through semantics and discourse.* **(GLM):** *Recurrent sequence motifs of a few nucleotides, regulatory element activity, and gene expression patterns.*

<u>Units of information</u>
   **(NLP):** *Fundamental units are letters, words, and sentences.* **(GLM):** *Core units are nucleotides (A, T, C, G), codons, motifs, and genes, often organized into regulatory or functional regions.*

<u>Contextual dependencies</u>
   **(NLP):** *Relies on short-range context (e.g., word dependencies) and long-range context (e.g., across sentences or paragraphs).* **(GLM):** *Short-range context involves local nucleotide motifs, long-range context includes distant regulatory elements interacting to affect gene expression.*

<u>Data variability</u>
   **(NLP):** *Highly variable, with semantic meaning unchanged via use of synonyms, rephrasing, and grammatical errors.* **(GLM):** *High genetic variability, including SNPs, short insertion-deletions, and structural variants, between individuals, with largely unclear consequences.*

---

a common occurence, being difficult to align). The authors aimed to address these drawbacks with PhyloGPN (Albors et al, 2025), and subsequently with GPN-Star (preprint: Ye et al, 2025).

Another approach relies on training on the human pan-genome graph constructed from 47 individuals from the Human Pangenome Reference Consortium (Liao et al, 2023), as done in DeepGene (Zhang et al, 2024a). During its training, DeepGene processes each subgraph as a separate training sample, serializing and processing it as DNA sequences of ~500 bps. Another gLM, BioFM (preprint: Medvedev et al, 2025), trained on the 1000 Genomes dataset (The 1000 Genomes Project Consortium, 2015), incorporates genetic variants as different letters (tokens) in its sequences, thus modeling small insertion-deletions (indels) and single-nucleotide polymorphisms (SNPs). In addition, it denotes the boundaries of coding/exonic regions and transcripts, injecting structural annotations directly into the genomic representations. The ablation study of BioFM suggests that both types of annotations contribute to BioFM's superior performance on variant-related prediction tasks (preprint: Medvedev et al, 2025).

### Models trained on transcribed regions

Other models focus only on certain regions of the genome. Species-LM offers models trained on different untranslated regions (UTR) of fungi and metazoa. By training only on DNA upstream (1 kb) and downstream (300 bps) of gene coding region boundaries, Species-LM managed to capture evolutionary conservation in UTR and promoter regions, recognizing regulatory elements like transcription factor and RNA binding motifs (Karollus et al, 2024, Tomaz da Silva et al, 2025). UTR-based gLMs on human genomes have also been developed (Chu et al, 2024; Yang et al,

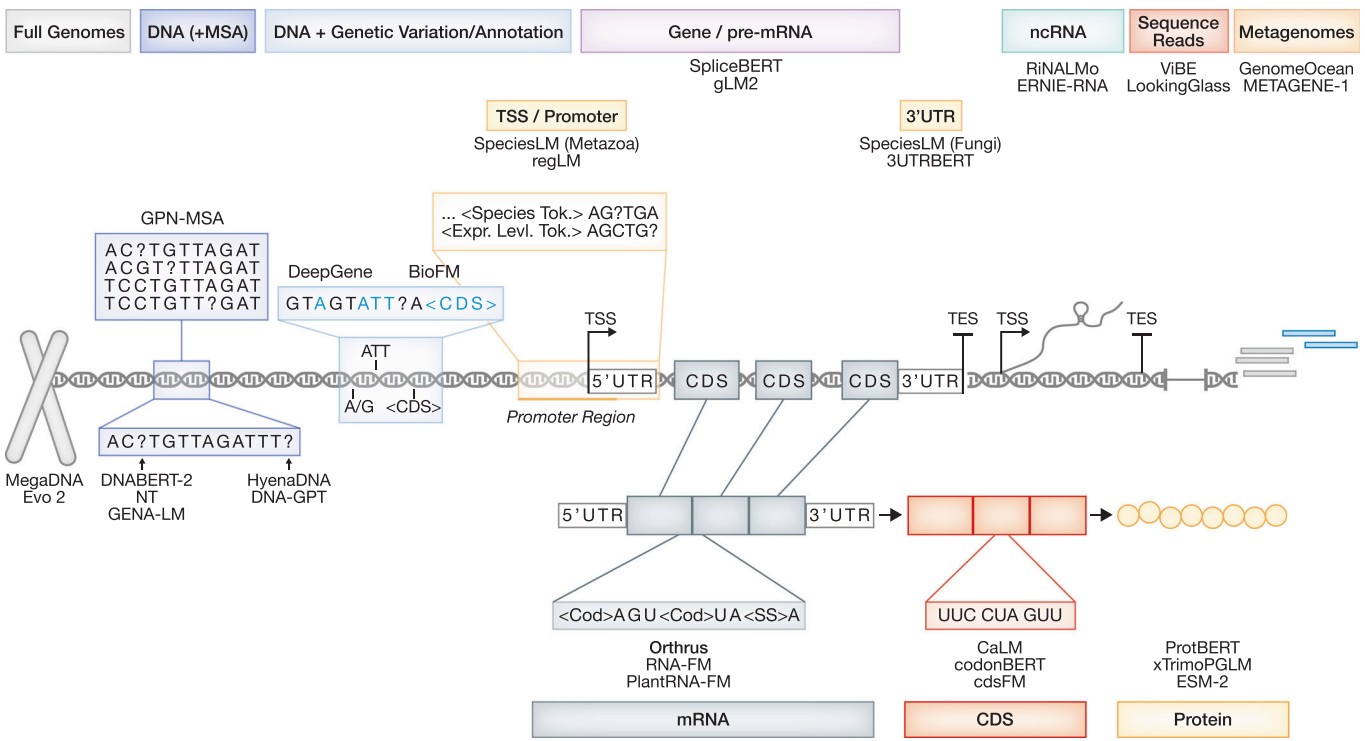

**Figure 1. Data modalities of genomic language models.**

Schematic of some of the different data modalities used to train genomic language models (gLMs). While some models learn on all genomic regions (left), others pretrain only on specific regions or modalities (right), for instance, on promoter regions, and coding sequences, giving them a predictive advantage in the corresponding domains. Most gLMs are pretrained to reconstruct the masked sequence denoted as "?". Special tokens (commonly marked with "<>") can be used to enhance sequence understanding, denoting species of origin, genomic annotation, or genetic variation. MSA multiple sequence alignment, TSS transcription start site, TES transcription end site, UTR untranslated region, ncRNA non-coding RNA, pre-mRNA precursor messenger RNA, mRNA messenger RNA, CDS coding DNA sequence, NT Nucleotide Transformer.

2025). Another model, regLM (Lal et al, 2024), focused on yeast promoter and human enhancer sequences. By incorporating the associated gene expression levels into the sequence (as tokens), regLM can be prompted to generate regulatory regions with desired sequences in a cell-type-specific manner.

GLMs trained on gene sequences have the potential to enhance several key downstream tasks, such as predicting gene mRNA expression, post-transcriptional modifications, mRNA properties, and splicing events, with the goal of outperforming domain-specific sequence-to-function models (preprint: Wang et al, 2025). Examples include SpliceBERT (Chen et al, 2024), an RNA model trained on pre-mRNA sequences of 72 vertebrate genomes, and Orthrus (Fradkin et al, 2024), trained on 45 M+ mature mRNA sequences of mammals. Orthrus has an efficient architecture, and thus can handle contexts of 12 kb; after splicing, this is enough to capture full mature transcripts, enabling Orthrus to outperform expert methods on mRNA property prediction tasks, even from very few data points.

Since mRNA transcripts are read by the translation machinery codon-by-codon, some models like cdsBERT (preprint: Hallee et al, 2023), CaLM (Outeiral and Deane, 2024), and codonBERT (Li et al, 2024b), process only the coding regions of transcripts in codons. These codon-based models learn the genetic code, and as such, are competitive both on mRNA and protein property prediction tasks, even outperforming some pLMs (Boshar et al, 2024). This suggests

that the representation learned by these models is complementary to that learned by pLMs, as they may capture global patterns in mRNA/protein sequences, but less so the finer-grained effects related to structure and inter-residue interactions.

RNA sequences fold into various shapes according to their function. Thus, similarly to pLMs, some RNA language models were trained with secondary structure prediction in mind, RNA-MSM (Zhang et al, 2024b), RiNALMo (Penić et al, 2025), ERNIE-RNA (Yin et al, 2025), or they were designed to generate novel RNA sequences that form stable secondary structures, e.g., GenerRNA (Zhao et al, 2024). For example, RiNALMo has been pretrained on 36 M non-coding RNA (ncRNA) sequences from RNAcentral (The RNAcentral Consortium et al, 2019), and other databases. It feeds learned nucleotide pairings into a folding head, giving accurate predictions for RNA secondary structure, even generalizing to new RNA families. Notably, the representations learned on ncRNA sequences can also be relevant for coding RNA, as RiNALMo outperforms SpliceBERT, and other RNA gLMs on splice site prediction and mean ribosome load prediction tasks.

### Mixed-modal and multimodal approaches

The central dogma of molecular biology states that genetic information flows from DNA to RNA to proteins. Recognizing that the same genetic information can be represented with different data modalities,

**Table 1. Influential genomic language models across different data modalities.**

| Source | Model (Family) | Modality | Training data | Application |
|---|---|---|---|---|
| (Schiff et al, 2024) | Caduceus | DNA | hg38 | - Long-range genomic tasks<br>- Regulatory variant effect prediction<br>- Splice site annotation |
| (preprint: Naghipourfar et al, 2024) | cdsFM | Codon | Bacterial + Eukaryotic cDNA | - Synonymous variant impact prediction<br>- Localization of open reading frames |
| (preprint: Brixi et al, 2025) | Evo 2 | DNA | OpenGenome2 (MS) | - Genome-wide variant effect prediction<br>- Genome-scale generation of multiple species<br>- Directed generation of sequences with desired epigenomic signals<br>- Multimodal generation tasks |
| (Fishman et al, 2025) | GENA-LM | DNA | T2T, 1000 G, Fly (ENSEMBL)<br>Yeast (ENSEMBL), Arabidopsis thaliana (ENSEMBL) | - Recognizing genomic elements<br>- Regulatory activity prediction |
| (preprint: Li et al, 2025) | Generanno | DNA | Prokaryotic genomes (RefSeq) | - Metagenomic annotation of sequences<br>- Gene fitness and pseudogene prediction<br>- Taxonomic classification |
| (preprint: Wu et al, 2025) | GENERator | DNA | Eukaryotic genomes (RefSeq) | - Protein-coding DNA generation and completion<br>- Regulatory sequence design<br>- Taxonomic classification |
| (Benegas et al, 2025a) | GPN-MSA | DNA - MSA | Conserved regions in humans with multiple sequence alignment of 100 vertebrate species | - Genome-wide variant effect prediction<br>- Variant classification |
| (Shao and Yan, 2024) | MegaDNA | DNA | Bacteriophage genomes | - Bacteriophage genome generation<br>- Virus classification<br>- Zero-shot gene essentiality prediction |
| (Liu et al, 2025b) | METAGENE-1 | Metagenomes | Wastewater metagenomic dataset | - Pathogen detection in wastewater samples<br>- Anomaly detection in metagenomic data |
| (Dalla-Torre et al, 2025) | NT | DNA | HG38, 1000 G, MS (RefSeq) | - Epigenetic mark prediction<br>- Recognizing genomic elements<br>- Regulatory activity prediction |
| (Fradkin et al, 2024) | Orthrus | RNA - 6OHE | Six-track mammalian mRNA sequences (Zoonomia) | - mRNA property prediction from few data points<br>- Functional clustering of splice isoforms |
| (Zhai et al, 2025) | PlantCaduceus | DNA | 16 angiosperm genomes | - Annotating newly sequenced Angiosperm genomes<br>- Cross-species evolutionary constraint prediction |
| (Penić et al, 2025) | RiNALMo | RNA | ncRNA sequences from multiple databases | - Inter-family RNA secondary structure prediction<br>- mRNA property prediction<br>- ncRNA family classification |
| (Karollus et al, 2024) | Species-LM | DNA | Fungi 5' UTRs, Fungi 3' UTRs | - Recognizing genomic elements<br>- Regulatory activity prediction<br>- Inferring RNA structure via Nucleotide Dependency maps |

mixed-modal approaches endeavor to combine these representations. For example, gLM2 (Cornman et al, 2024) was trained on metagenomic (primarily prokaryotic) data of the OMG dataset (Cornman et al, 2024), translating coding regions to amino acid tokens, while representing non-coding regions as nucleotides. Thereby, gLM2 can compete in both protein and DNA benchmarks. Additional mixed-modal gLMs have been trained to enable crosstalk between DNA-RNA-protein (He et al, 2025, preprint: Zhu et al, 2024), DNA-epigenetics (Trotter et al, 2021), and natural language (de Almeida et al, 2025a, 2025b), with some reported benefits to model performance.

In contrast, multimodal approaches integrate the already processed representations of different modalities, often generated by other pretrained models. For instance, Isoformer (Garau-Luis et al, 2024) combines representations of DNA from Enformer, RNA from NT, and proteins from ESM2, achieving state-of-the-art performance in transcript isoform prediction. Their analysis shows that all three representations contribute meaningfully to the task. Building on this, an interesting new paradigm of natural language and genomic multimodal approaches has appeared, which combines the knowledge from DNA foundation models with the reasoning capabilities of LLMs to enable deeper biological

**Table 2. Design choices and architectures of influential genomic language models.**

| Model (Family) | Context size | Tokenizer | Objective | Architecture | Model size |
|---|---|---|---|---|---|
| Caduceus | 131 Kb | Nucleotide | MLM | Decoder - SSM - BiMamba | 0.5 M, 2 M, 8 M |
| cdsFM | 2048 codons | Codon | MLM, CLM | Encoder, Decoder - Transformers - RoFormer | 80 M, 200 M, 620 M, 1 B |
| Evo 2 | 1 Mb | Nucleotide | CLM | Decoder - Hybrid - Striped Hyena 2 | 7 B, 40 B |
| GENA-LM | 4.5 Kb, 36 Kb | BPE | MLM | Encoder - Transformers – BERT, BigBird, Recurrent Memory Transformer | 110 M, 336 M |
| Generanno | 8 Kb | Nucleotide | MLM | Encoder - Transformers - Llama | 500 M |
| GENERator | 98 Kb | Non-overlapping 6-mer | CLM | Decoder - Transformers - Llama | 0.5 B, 1.2 B, 3 B |
| GPN-MSA | 128 nt | Nucleotide | MLM | Encoder - Transformers - RoFormer | 86 M |
| MegaDNA | 96 Kb | Nucleotide | CLM | Decoder - Transformers - MEGABYTE | 78 M, 145 M, 277 M |
| METAGENE-1 | 512 tokens | BPE | CLM | Decoder - Transformers – Llama 2 | 7 B |
| NT | 6 KB, 12 Kb | Non-overlapping 6-mer | MLM | Encoder - Transformers – BERT, RoFormer | 50 M, 500 M, 100 M, 250 M, 2.5 B |
| Orthrus | 12 Kb | Nucleotide | CL | Encoder - SSM - Mamba | 1 M, 10 M |
| PlantCaduceus | 512 nt | Nucleotide | MLM | Decoder - SSM - BiMamba | 20 M, 40 M, 225 M |
| RiNALMo | 1022 nt | Nucleotide | MLM | Encoder - Transformer - RoFormer | 650 M |
| Species-LM | 2 Kb | Nucleotide | MLM | Encoder - Transformers – BERT, RoFormer | 90 M |

understanding. Approaches like ChatNT (de Almeida et al, 2025b), and BioReason (preprint: Fallahpour et al, 2025) process biological sequences via gLMs and integrate the resulting representations with English textual queries, allowing them to reason about genomic information, generating human-readable explanations and predictions for complex biological problems. Ultimately, the ability to translate between different modalities increases input diversity and often leads to more informed representations, yet requiring all modalities at inference time can limit the applicability of the model.

### Data curation and generalization trade-offs

Beyond modality, the curation of genomic data strongly influences gLM performance. A central challenge is the treatment of repetitive and intergenic regions: repetitive sequences, common in eukaryotes including plants (Zhai et al, 2025), are easy for language modeling yet contribute less functional signal. Some observations indicate that including intergenic data in the pretraining tasks can reduce downstream utility (Ellington et al, 2024; preprint: Wu et al, 2025). Mitigation strategies include focusing on functionally relevant or conserved regions (Benegas et al, 2025a), and down-weighting repetitive content (Ellington et al, 2024).

While models trained on diverse, multispecies datasets aim to leverage evolutionary information for broad generalization (Dalla-Torre et al, 2025), maximizing phylogenetic diversity is not universally optimal. Distinct genomic features may require specific evolutionary scopes: while deep timescales are helpful for coding and rare variants, functional non-coding and common variants are better captured at shallower timescales (preprint: Ye et al, 2025). Reflecting this need for specialization, region- or taxon-specific models can yield superior performance when sufficient pretraining data is available (Karollus et al, 2024, preprint: Benegas et al, 2025b, Fishman et al, 2025). Domain adaptation (see below) or continual pretraining may be used when such data is scarce or of low quality (preprint: Baghbanzadeh et al, 2025).

## Tokenization strategies

### Fixed and dynamic K-mer tokenizers

Tokenization (depicted in Fig. 2A) is a crucial step in language modeling, defining how input sequences are represented within the model. Unlike natural language, however, DNA and RNA are not naturally structured into well-defined units of words or sentences and has a small vocabulary of only four letters (A, C, G, T/U), which encode often overlapping regulatory features (Lindsey et al, 2024). A common method is to treat contiguous k-length segments of nucleotides (k-mers) as tokens (Ji et al, 2021), with different values for k ranging from 3 to 6. If these k-mers are overlapping with a stride of 1 (shift between tokens), as in the case of DNABERT, we require roughly the same number of tokens to represent a sequence, as the number of nucleotides it contains. Moreover, neighboring tokens may leak information during standard NLP pretraining tasks, leading to the model memorizing sequence identity (Sanabria et al, 2024b). More often, DNA sequences are tokenized into non-overlapping k-mers, with a stride equaling k, to reduce sequence length by a factor of k (Dalla-Torre et al, 2025). NT uses six-mers, which performed best among the tested token lengths. However, compression alone has little impact on accuracy (Lindsey et al, 2024), and fixed k-mers may not align with meaningful units like codons (Boshar et al, 2024), potentially limiting gLMs' contextual understanding. A recent NT version showed that codon-aligned three-mers improve protein-related predictions compared to six-mers (Lindsey et al, 2024).

Byte pair encoding (BPE) (Sennrich et al, 2016) is a method adapted from NLP and offers a more targeted vocabulary by iteratively merging frequently co-occurring pairs of tokens, used in DNABERT-2 (Zhou et al, 2024). Comparative studies of BPE against non-overlapping k-mer representations concluded that BPE resulted in lower pretraining loss, faster convergence (preprint: Baghbanzadeh et al, 2025) and produced significant performance gains on downstream tasks (Zhou et al, 2024). The vocabulary size

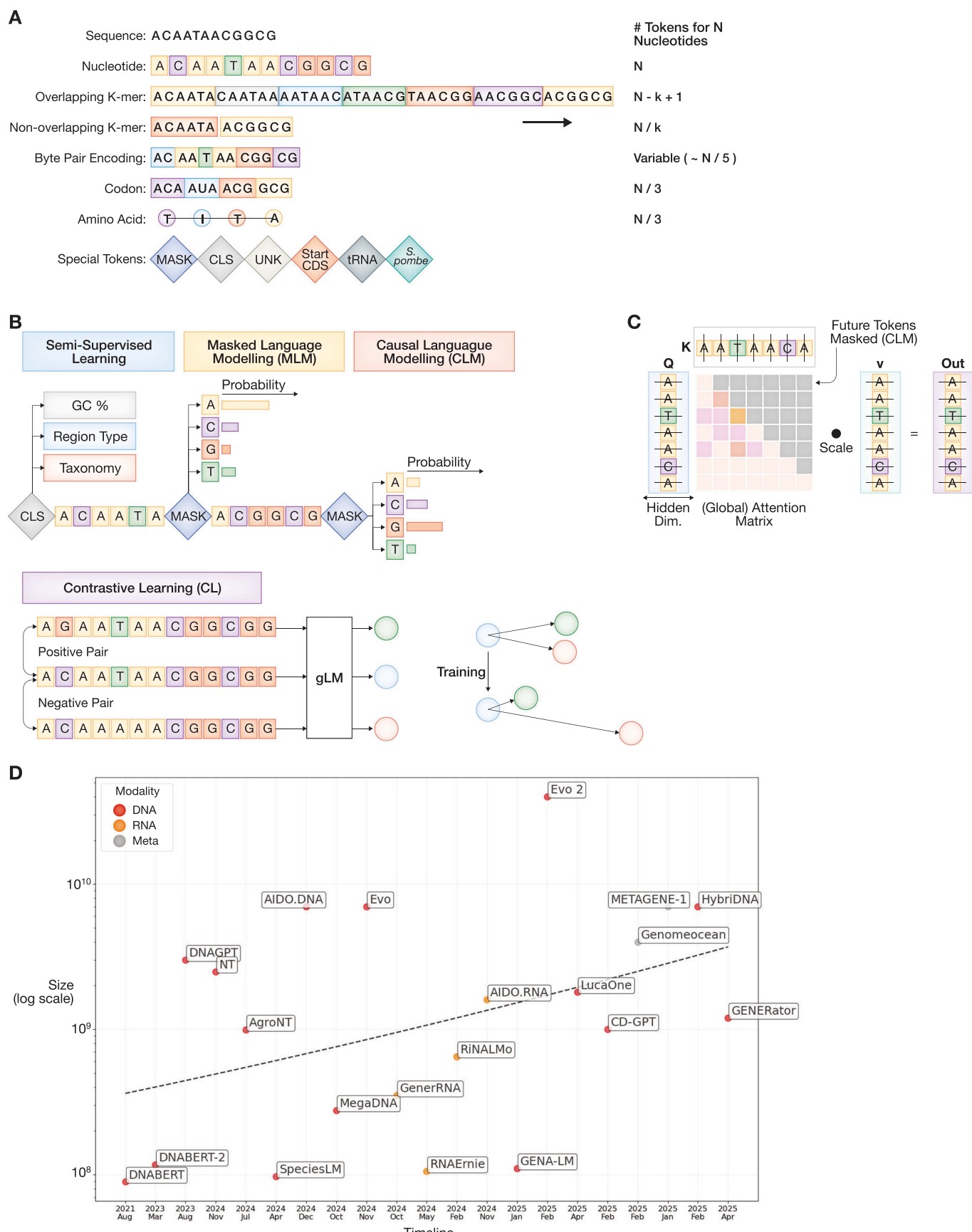

**Figure 2.   Components of genomic language models and their evolution.**

(A) Schematic depicting common ways of tokenizing a sequence, their corresponding compression rate, and some special tokens commonly used. (B) Typical pretraining strategies for gLMs include reconstructing masked tokens in the middle (MLM), or at the end of sequences (CLM) from the token with the highest predicted probability. Contrastive learning, on the other hand, trains the model to distinguish between similar and dissimilar sequence pairs, encouraging representations that capture meaningful differences. Semi-supervised learning objectives (for example, species of origin prediction) are sometimes used to enhance the learned representations. (C) Attention scores of a decoder gLM across input nucleotides, revealing how the model distributes focus when generating each token. Higher attention weights along the diagonal indicate a strong reliance on recent context, while off-diagonal patterns highlight longer-range dependencies learned during training. These values are used to weigh the representations of each nucleotide. In this case, the model is trained via CLM, and thus, token identity is only affected unidirectionally from the left. (D) Since their appearance (time on x-axis), gLMs have been growing in size (y-axis, log-scale) across each data modality (here DNA, RNA, Metagenomes; shown by color), reaching over 1 billion parameters. Only selected gLMs, and only the largest instance of each model family is plotted.

of BPE, determined by the number of token merges, has a large impact on the downstream performance of the model, with smaller vocabularies generalizing better. To avoid biasing the model towards specific biological tasks, GROVER uses an intrinsic validation approach to determine the optimal vocabulary size, namely, next k-mer token prediction, ensuring that all the tokens are relevant to represent different genomic content. As a result, GROVER outperformed DNABERT-2 on promoter recognition tasks (Sanabria et al, 2024a). In addition, domain-specific BPE tokenizers might show variable overlap with vocabularies obtained from the pretraining set (Lindsey et al, 2024), suggesting, for example, that not all tokens learned on multispecies datasets are equally useful in species-specific downstream tasks, and that BPEs with lower vocabularies tend to generalize better (preprint: Baghbanzadeh et al, 2025). BPE can also compress the input sequence significantly, although its compression rate depends on the vocabulary size, and whether the input sequence is primarily represented by more or less frequent tokens.

### Single nucleotide and control tokens

Small changes in gLM input sequences, such as single-nucleotide variants (SNVs) and small insertion-deletion variants (indels), can lead to dramatic differences in the tokenized representation (Patel et al, 2024; preprint: Baghbanzadeh et al, 2025). As a consequence, BPE, or non-overlapping k-mers, are not a convenient choice for modeling variant effects and genetic variation. Moreover, it has been suggested that a consistent representation of similar sequences may ease convergence during pretraining, and it has been shown to have a positive effect on downstream performance (Lindsey et al, 2024).

The need to compress sequences into shorter representations often stems from architectural constraints, especially since even distal regulatory elements can significantly affect specific DNA regions. In contrast, protein sequences are much shorter, and tokens typically represent individual residues. Recently developed gLM approaches (Fradkin et al, 2024; Nguyen et al, 2023; Nguyen et al, 2024) alleviate the concern of limited context windows and can operate at base-pair resolution even across very large contexts of up to 1 Mb, without significantly increasing computational costs. In these models, single-nucleotide tokens have been shown to outperform BPE and k-mers (Lindsey et al, 2024), and they may enable more favorable scaling between model complexity and performance (Nguyen et al, 2024; preprint: Brixi et al, 2025).

Most gLMs introduce special control tokens that aid the modeling process and prediction tasks. Since the models produce embeddings for each token in the sequence, it is typical to include an artificial CLS (classification) token at the beginning of sequences, which is incentivized to have a global representation over all tokens, and is often used in downstream prediction tasks. Similarly, a SEP (separation) token is often appended at the end of sequences, which is sometimes useful if pairs of input sequences are processed at once (Li et al, 2024a). In addition, as we have seen, some models include tokens with biological relevance, denoting biotype, species (Karollus et al, 2024), strandedness, expression values (Zhang et al, 2023) or even prompts in natural language (preprint: Fallahpour et al, 2025), which can influence representations, and guide generation tasks (Lal et al, 2024). All of these included features affect model performance or applicability and need to be selected carefully during model development.

## Learning objectives

### *Self-supervision via token prediction or contrastive learning*

The transfer learning paradigm aims to utilize a model trained to perform a pretraining task to support the development of a model intended for a targeted downstream application. A simple pretraining objective that may provide useful information for genome interpretation and understanding is language modeling on DNA/RNA sequences, namely, predicting which nucleotides are more likely to occur in the context of other given nucleotides. Since genomes are under evolutionary constraints, the syntax learned by gLMs should enable them to recognize various functionally important elements. GLMs trained through masked language modeling (MLM) (see Fig. 2B) learn to predict the identity of contiguous masked tokens (commonly 10–20%), from their surroundings. Since these models (Dalla-Torre et al, 2025; Fishman et al, 2025; Ji et al, 2021) aggregate information from both directions of the masked tokens, they tend to achieve representations with richer semantic understanding and transfer learning capabilities, than models trained in an autoregressive manner, also known as causal language modeling (CLM). In CLM, only the last token of a sequence is masked, and thus, CLM aggregates information unidirectionally, making it a common choice for generative models, e.g., (Lal et al, 2024; Nguyen et al, 2024).

Contrastive learning (CL) is another method being explored for pretraining gLMs. The core idea of CL is to teach the model a structured latent space, by pulling "positive" pairs of samples together (samples that have similar biological effect), while pushing "negative" pairs apart (those with different biological implications). This pushing and pulling (depicted in Fig. 2B) is achieved via similarity measures between the embedding vectors of samples. DNABERT-S (Zhou et al, 2025b) employs this strategy to further train the DNABERT-2 model, by matching DNA sequences from

different organisms. If two pieces of non-overlapping DNA sequence are from the same species, they are considered a positive pair, and negative if they come from distinct ones. This learning strategy led to significant improvements on Genome Understanding Evaluation (GUE) benchmark datasets (Zhou et al, 2024), while also improving DNABERT-S in metagenomics binning.

Following a similar approach, Orthrus (Fradkin et al, 2024) matches alternatively spliced isoforms of transcripts, as well as homologs of the same transcript in different organisms. The hypothesis is that these evolutionarily related pairs are more functionally similar than randomly sampled RNA sequences. Orthrus' ability to encode these invariances helps it learn complex mRNA properties, outperforming other gLMs and expert methods, especially in the case when few training samples are available. CL has also been tested for non-coding variant effect prediction, with the framework DYNA (Zhan et al, 2025), in which different gLMs (and pLMs) were fine-tuned to maximize embedding similarities between wild type and benign alternative sequences, while minimizing it between wild type and pathogenic variants. In another study, the developers of UKBioBERT (preprint: Liu et al, 2025c) found MLM to lead to better performance than CL during pretraining; however this analysis did not consider additional CL strategies. The effect of integrating both types of learning objectives (CL + MLM) has since been tested, proving that this is an effective way of boosting predictive performance (preprint: Dalal et al, 2025).

### Improving understanding via auxiliary supervised tasks

Similar to NLP, some gLMs enhance the learning process by introducing additional pretraining tasks, in a self-supervised or often in a supervised manner. DNAGPT (Zhang et al, 2023) predicts the GC content, and sequence order (i.e., whether segments of a contiguous DNA sequence have been rearranged or not), on top of CLM, although it was less well established whether these prediction tasks contributed to improved performance. CD-GPT (preprint: Zhu et al, 2024) is a mixed-modal gLM accepting both nucleotide and peptide sequences as input. It was trained in separate phases, first with CLM, then by distinguishing between matching mRNA and protein sequences, the model learns the rules of translation and codon language. LucaOne (He et al, 2025) applies an even more extensive pretraining strategy, with eight additional supervised pretraining tasks based on genomic annotations. These tasks fall under categories such as span-level classification (e.g., recognizing types of genomic regions), structure-level regression (e.g., tertiary structure prediction), and seq-level classification (e.g., taxonomy prediction). Structure-based supervised pretraining tasks are also common for RNA-based gLMs. RNABERT used the token embeddings to predict correct structural alignment of the RNA sequence (Akiyama and Sakakibara, 2022), and both PlantRNA-FM (Yu et al, 2024), and UTR-LM (Chu et al, 2024) pretraining include secondary structure tasks.

## Architectures

### Pre-transformer architectures

Prior to the widespread adoption of the transformer architectures, convolutional neural networks (CNNs) and recurrent neural networks (RNNs), or CNN-RNN hybrids, were typical choices for sequence-to-function models (Sokolova et al, 2024). 1D convolutions are particularly effective at capturing local patterns such as sequence motifs, and CNN baseline architectures such as ResNet (He et al, 2016) and UNet (Ronneberger et al, 2015) are relevant to this day for benchmarking. Convolutional layers can also be stacked to encode input sequences, as in GPN (Benegas et al, 2023), and GPN-Promoter (preprint: Benegas et al, 2025b), two CNN-based gLMs trained via MLM that showed promising results on zero-shot variant effect prediction in Brassicales genomes and promoter regions of animals, respectively. CNNs perform well on structured data like images or aligned sequences. For modeling language, RNNs were the standard architecture prior to Transformers, as they focus on the sequential nature of language, maintaining a form of memory or state as they process inputs of varying length (Consens et al, 2025a). This is achieved by utilizing recurrent connections, whereby the output of a neuron at one time step is fed back to itself in the next time step. A variant of RNNs, called long short-term memory (LSTM) was used in the model Self-GenomeNet (Gündüz et al, 2023), and in LookingGlass (Hoarfrost et al, 2022). That model was pretrained with CLM on short microbial DNA reads, and it was applied to downstream metagenomic tasks (Hoarfrost et al, 2022).

### Transformers and variants

Nonetheless, RNNs are difficult to scale up to larger model sizes, as they cannot be parallelized. Thus, most discussed gLMs rely on the transformer architecture, characterized by the self-attention mechanism (Vaswani et al, 2017). This process allows for the crosstalk between every pair of tokens in the sequence (Fig. 2C), while also supporting parallel computation. It is common to differentiate between encoder-only, and decoder-only frameworks, which in practice often corresponds to using MLM or CLM for pretraining, respectively, with the appropriate attention masking strategy.

Many gLMs follow the encoder-only architecture developed by Google, BERT, for example (Ji et al, 2021). The original BERT model uses 12 multi-headed attention (MHA) layers of hidden dimension 768, resulting in around 110 M trainable parameters (Devlin et al, 2019). It is common practice to adapt the BERT architecture, by reducing (or increasing) the number of layers and hidden dimension based on computational and performance needs, e.g., SpliceBERT only uses six layers of dimension 512, thus reducing the number of parameters to a more manageable 19 M (Chen et al, 2024). Later gLMs are based on successors of BERT, such as RoBERTa (preprint: Liu et al, 2019) e.g., FloraBERT (Levy et al, 2022), ALBERT (preprint: Lan et al, 2020) e.g., LOGO (Yang et al, 2022), and DeBERTa (preprint: He et al, 2021) e.g., seqLens (preprint: Baghbanzadeh et al, 2025). These variants mostly reduced model size and inference time through optimization tricks, while maintaining model performance (in NLP tasks).

Decoder-only transformers mostly rely on the GPT (preprint: Radford et al, 2018) architecture, e.g., DNAGPT, Llama (preprint: Touvron et al, 2023), e.g., METAGENE-1 (Liu et al, 2025b), or on Mistral (preprint: Jiang et al, 2023), e.g., GenomeOcean (preprint: Zhou et al, 2025a). However, the largest constraint of (encoder or decoder) transformer-based architectures is their limited context size.

On one hand, this constraint stems from that BERT-based models are made position-aware via fixed-length positional encodings (the pairwise nature of the attention mechanism ignores positional information). Many gLMs thus rely on Rotary Positional

Embeddings (RoPE for short) introduced in RoFormer (Su et al, 2024), as used in NT, AIDO.DNA, GENA-LM. RoPE allows the network to generalize to sequences longer than those it was pretrained on, and is the easiest off-the-shelf solution, as encodings are simply added to the token embeddings. Other positional encoding schemes, such as Attention with Linear Biases (ALiBi) (Press et al, 2021) and relative positional encoding, used in T5 (Raffel et al, 2020). On the other hand, computing the full attention in every layer is costly, further constraining input size, which is why most transformer-based gLMs only process inputs of up to ~10 kb (see Dataset EV1). Thus, recent advancements focused on reducing computational overhead via omitting the attention computation for some pairs of tokens, for example (Zaheer et al, 2020), or leveraging hardware-aware algorithms for attention calculation, namely FlashAttention (Dao et al, 2022).

### The post-transformer era

Transformer-based architectures still suffer from computational complexity, which scales superlinearly with sequence length, making accommodation of long sequences challenging (Consens et al, 2025a). State-space models (SSMs) (Fu et al, 2023) can be interpreted as RNNs that bypass the temporal dependencies via convolution operators, thus processing the whole sequence in parallel, with (near-)linear scaling. The first of these architectures to be adapted to DNA, was HyenaDNA, relying on the Hyena operator (Nguyen et al, 2023) instead of multi-head attention; this allowed processing 1 Mb of DNA in one pass, making it adept at ultralong-range genomic prediction tasks. Orthrus (Fradkin et al, 2024), and Caduceus (Schiff et al, 2024) are two more compact-sized SSMs (less than 10 M parameters). Caduceus uses a Reverse complementary equivariant Mamba (Gu and Dao 2024), a module that processes both strands of DNA simultaneously. This model has a supported context size of 131 kb and can often be found among the top performers on benchmarks, even though it has only been pretrained on a human reference. Hybrid models like Evo (Nguyen et al, 2024) and Evo 2 (preprint: Brixi et al, 2025), with alternating Hyena-Transformer blocks, and HybriDNA (Ma et al, 2025), with alternating Mamba2-Transformer blocks, attempt to combine the benefits of both architectures, with improved scalability and no loss of performance due to compression. As a result, much of the current progress centers on refining these hybrid architectures, which seek to synergize the global context modeling of transformers and the scalability of state-space models. Taken together, current results suggest that future best-performing gLMs are unlikely to be pure Transformers, but will increasingly adopt hybrid designs that combine attention and/or local convolution with state-space modules. We anticipate a division of labor: encoder-only Transformers will likely remain the default backbone for compact, task-focused gLMs, while large generalist and design-oriented models will converge on hybrid, potentially SSM-heavy stacks that scale quasi-linearly with sequence length.

## Scalability and model complexity

### Scaling laws of LLMs

We note an increasing trend in model complexity and pretraining dataset sizes (Fig. 2D; Table EV1). Whereas DNABERT with 89 million parameters was of comparable size to previous state-of-the-art sequence-to-function models, like Enformer (Avsec et al, 2021)

~250 M and Borzoi (Linder et al, 2025) ~ 186 M, there are now several gLMs with parameter counts over 1 billion (see Dataset EV1). At the current time, Evo 2 is the largest pretrained model available with a 40 B parameter count, although this is still less than half of the largest pLMs, such as ESM3 (Hayes et al, 2025), xTrimoPGLM (Chen et al, 2025), and 1–2 magnitudes smaller than the most recent models in NLP (preprint: Achiam et al, 2024; Yang et al, 2025).

In accordance with the scaling laws seen in NLP (preprint: Kaplan et al, 2020), several studies have observed consistent improvements in model performance increasing with complexity across different architectures (preprint: Liu et al, 2024, Dalla-Torre et al, 2025; Ma et al, 2025; Nguyen et al, 2023), but with significant differences (preprint: Brixi et al, 2025). Configurations with lower pretraining loss consistently achieved better results on downstream tasks. However, better performance has also been attributed to longer context size (Liu et al, 2024; Ma et al, 2025), consequently, even though SSMs usually lag behind large transformer-based models on shorter context sizes, they perform better on long-range genomic tasks (Liu et al, 2024), where context is not limited. LLM scaling laws also state that, alongside model complexity, dataset size and compute time also require scaling up to achieve optimal performance (preprint: Kaplan et al, 2020). These relationships inform the required pretraining dataset size of a given model configuration, if trained to convergence. For example, according to some estimates, at least 20 B tokens are required for pretraining a model with 1 B parameters, with an increasing number of tokens for each additional billion parameters (preprint: Kaplan et al, 2020; Nguyen et al, 2024).

### Will we hit a scaling wall?

It follows from established scaling laws that exponentially increasing resource requirements tend to yield diminishing returns. In the case of pLMs, a scaling wall has been observed in zero-shot settings, where models larger than 1–4 billion parameters no longer show consistent performance gains (Notin et al, 2023). This aligns with estimates of the parameter scale needed for protein folding tasks, assuming that pLMs primarily learn evolutionary motif couplings rather than a full biophysical model of folding (Zhang et al, 2024c).

However, these findings contrast with those from (Chen et al, 2025), where the 100-billion-parameter model achieved the best results in both probing and fine-tuning approaches, and to our knowledge, no comparable findings have been reported for gLMs, and scaling laws for these models are yet to be studied in detail. We emphasize that model architecture and input modality likely play more significant roles in downstream performance than model size alone, and currently, no single model consistently outperforms all others across prediction tasks (preprint: Wang et al, 2025).

Lastly, we note that training gLMs of large sizes requires specialized software to split work across GPUs, and models larger than 100 M are infeasible to train without a cluster of GPUs. The amount of training time also varies significantly, taking weeks to months until convergence, and incurring significant operational costs. Some estimates based on PFS-Days (Amodei and Hernandez, 2018) put the price of pretraining a >100 M parameter gLM at several thousand USD as a baseline (Consens et al, 2025a). Simply utilizing the already trained models might also require access to GPUs, depending on the application and model, further limiting

accessibility for research labs with less computational capacity or budget.

# Using and evaluating genomic language models

We have introduced a large number of methods with different architectural designs and pretraining strategies. In this section, we discuss how gLMs can be used for prediction tasks, and how they are evaluated. Many competing gLMs appeared over the recent years, and a number of independent benchmark datasets have been proposed, similar to NLP. Here, we collect the most prominent benchmarks (reported as an independent publication, or frequently used ones) and summarize key findings. Subsequently, we discuss the challenges most current gLMs face.

## Using genomic language models for prediction tasks

### Zero-shot prediction

Once pretrained, gLMs can be applied to a variety of downstream tasks via transfer learning techniques. The simplest approach, zero-shot prediction, in NLP context refers to the ability of LLMs to perform tasks not encountered during pretraining with no additional learning. In the genomics context, this is often used to assess the deleteriousness of functional variants in the genome. Logits from models pretrained via MLM or CLM can be interpreted to represent the likelihood of each alternative allele being observed in a healthy population at a specific position. Thus the log-likelihood ratio between reference and alternative alleles is often used as a zero-shot score for measuring variant effects (Benegas et al, 2025c; Benegas et al, 2023), and has been shown to reflect learned functional constraints across different modalities, e.g. transcription factor binding sites in DNA (Benegas et al, 2023), splicing (Chen et al, 2024), non-coding variants (Zhan et al, 2025), coding mutations as amino acids (Rives et al, 2021; Zhan et al, 2025) and DNA (preprint: Naghipourfar et al, 2024). This approach is more suitable for gLMs trained at single-nucleotide resolution, rather than fixed k-mers (or BPE), especially if indels are also considered, which may change the splitting of the tokenized sequence drastically. Such variants can be easily interpreted via CLM-based models or relying on the pseudo-log-likelihood ratio with MLM-based ones (Rives et al, 2021).

Alternatively, as sequences with deleterious variants will likely be out-of-distribution compared to natural ones, distance metrics between the embedding vectors for the reference and alternative sequences can be used. For example, the cosine similarity between the embeddings at the position of the variant (preprint: Naghi-pourfar et al, 2024), or aggregated across the sequence (Patel et al, 2024), with the latter also suitable for models trained with BPE.

### In-context learning and probing of embeddings

Recent advances in NLP have also demonstrated that LLMs can adapt their behaviors to new tasks, in context, that is, from well-engineered prompts (Dong et al, 2024). In this case, information about the predictive task, such as examples of respective sequences and labels, are added to the input prompt of the model. By conditioning their prediction on the provided context, LLMs learn to perform the task without any changes to their parameters.

However, encoding labels via the limited vocabulary of gLMs is not straightforward. Two variants of in-context learning have been used to adapt HyenaDNA to downstream tasks by repurposing its limited existing vocabulary for classification (Nguyen et al, 2023). One approach, called few-shot learning, involves prepending a small number of labeled examples to the input. In contrast, soft prompt tuning utilizes learnable tokens injected directly into the input sequence, with only these tokens being optimized. Adding label tokens to the vocabulary of the model can enable generation conditioned on the control token (label), for example, guiding the generation to specific CRISPR-Cas system types (Nguyen et al, 2024), or generating promoter sequences that increase expression (Lal et al, 2024).

GLMs may also be used as feature extractors, as their latent vector representations potentially contain context-aware information relevant for the downstream learning task. As an example, many studies have demonstrated that embedding vectors form distinct clusters based on the class of genomic elements they contain, learned simply from the self-supervised pretraining task (Dalla-Torre et al, 2025; Benegas et al, 2023; Nguyen et al, 2023). The embedding vectors usually have dimensions between 256 and 8192, per token (preprint: Benegas et al, 2025b). This matrix may either be fed directly into a CNN or is often aggregated along the sequence length via mean pooling or max pooling. This can lead to better results more often than the values along the [CLS] token (preprint: Baghbanzadeh et al, 2025). In either case, the embeddings are used to train a simple machine learning algorithm to give predictions, called (linear) probing, depending on whether a linear or nonlinear model is used; in either case, the gLM parameters remain unchanged. These intermediate representations from a model may also change based on which layer they are obtained from (Dalla-Torre et al, 2025). Embeddings from initial layers consistently demonstrate suboptimal predictive performance, as they likely capture low-level sequence features. Similarly, final layers can exhibit a performance dip as they become more specialized in the pretraining task, thus best results are often achieved with intermediate layer representations (Dalla-Torre et al, 2025; Tang et al, 2025, Fishman et al, 2025). Combining representations from different layers can further boost predictive performance, albeit usually by a small margin (preprint: Baghban-zadeh et al, 2025).

Lastly, the representations learned by one model can also be taught to another one, thus enabling transfer learning via student-teacher knowledge distillation, as cdsBERT (preprint: Hallee, et al, 2023) implemented to align codon and amino acid token representations. Alternatively, knowledge distillation can also be used to transfer representations from a large, computationally expensive model to a smaller, more efficient one, enabling faster inference while retaining much of the performance.

### Full and parameter-efficient fine-tuning

The previously mentioned methods did not require changing the learned weights of gLMs. In contrast, during fine-tuning, a predictive head (dense neural net or CNN) is added, and then trained together with the base gLM for a specific downstream task. In general, performance tends to be superior via fine-tuning than probing (Dalla-Torre et al, 2025; preprint: Wang et al, 2023; Patel et al, 2024). Performance may further be boosted by additional pretraining on the domain-specific datasets (target genome, or

target dataset) with the original language modeling objective, a technique called domain adaptation (preprint: Baghbanzadeh et al, 2025; Zhou et al, 2024), although models trained this way can be prone to overfitting to the target task.

However, as gradient computations for gLMs require substantially more memory than required at inference time, full fine-tuning can be out of reach for many researchers. Recently, Parameter-Efficient Fine-Tuning methods (PEFT), such as Low-Rank Adaptation (LoRA) (Hu et al, 2022), have been proposed, which alleviate the resource needs of training by significantly reducing the number of parameters that get updated. This can be combined with quantization, e.g., QLoRA (Dettmers et al, 2023), implying a mapping of weights to lower-precision data types while fine-tuning, reducing computational and memory requirements. Ideal results via PEFT, however, depend on selecting the optimal hyperparameters, which could explain why training with LoRA often underperforms full-scale fine-tuning (preprint: Baghbanzadeh et al, 2025). It has also been observed that outlier samples can hinder the robustness of quantization, and that replacing vanilla attention layers with outlier-free layers, can increase fine-tuning performance, as in GERM (Luo et al, 2025a). This approach may also be applied to already pretrained models to improve the robustness of quantization via small-step continual learning (Luo et al, 2025a).

### Practical considerations

The barrier to entry for deploying gLMs has been significantly lowered by the standardization of model repositories. Most current architectures are readily available via platforms like Hugging Face, allowing researchers to perform inference and training using standardized pipelines, facilitating a smooth start. The computational burden can be further managed by the above-mentioned PEFT strategies; this approach enables the fine-tuning of substantial models on consumer-grade hardware rather than requiring datacenter-scale infrastructure.

Selecting the appropriate gLM requires balancing the scope of the biological inquiry against computational resources available. Extensive generalist architectures, such as Evo 2, capture diverse genomic patterns, but impose high resource demands. In contrast, domain-focused models like Species-LM or Orthrus prioritize efficiency, but have architectural priors tailored to specific biological properties. These specialized models often yield competitive performance on targeted downstream tasks with considerably greater computational efficiency, making them highly suitable for applied research settings.

### Interpretability techniques

Several techniques have been proposed to investigate the internal workings of gLMs, commonly starting with an unsupervised clustering of sequence embeddings to assess whether the model's learned representations align with known biological signals. These analyses frequently reveal clustering patterns that correspond to chromatin states, replication timing, and functional annotations (Nguyen et al, 2023; Dalla-Torre et al, 2025; Sanabria et al, 2024a; Chen et al, 2024).

Beyond static embeddings, the attention weights of transformer-based models can be visualized to highlight interactions between genomic regions, such as those between splicing donor and acceptor sites (Chen et al, 2024) or distinct regulatory elements (Dalla-Torre et al, 2025). We note, however, that transformers

contain multiple attention matrices (one for each head across multiple layers) and so visualizing individual attention heads provides an incomplete picture. Researchers can also derive importance scores by masking input nucleotides and assigning weights based on the model's predicted likelihood of the original base, as seen in GPN.

Perturbation-based approaches extend this idea by analyzing how changes in the input affect the output. Since gLMs output probability distributions, introducing point mutations at a specific position leads to shifts in nucleotide probabilities at other positions. Nucleotide dependency analysis utilizes this phenomenon to highlight important interactions between transcription factor binding sites and structural elements in RNA (Tomaz da Silva et al, 2025). A related approach, the categorical Jacobian (Zhang et al, 2024c), considers the sensitivity of the entire output logit matrix to mutations in the input sequence. While originally used for contact map prediction in pLMs (Zhang et al, 2024c), this method has recently been applied to analyze coevolutionary signals in gLMs (Nguyen et al, 2024).

These techniques are conceptually similar to "in silico mutagenesis" (ISM), often used to interrogate the supervised, sequence-to-function DNA models (Novakovsky et al, 2023). In ISM, each possible point mutation is given a score, based on the difference in predicted signals, a computationally intensive process (Sasse et al, 2024). The resulting "attribution map" is well-suited for finding important sequence elements de novo, for example, by integrating with TF-MoDISco (preprint: Shrikumar et al, 2018). Recent methods such as SQUID (Seitz et al, 2024), and its successor SEAM (preprint: Seitz et al, 2025) offer an end-to-end pipeline for interpretability. These methods are also applicable to unsupervised gLMs in principle; however, they would require constructing an attribution map based on gLM outputs.

Finally, the most recent trend involves adapting mechanistic interpretability techniques from NLP (preprint: Cunningham et al, 2023) by using sparse autoencoders to decompose dense model representations. This method has been applied to Evo 2 to isolate specific directions in the latent space that correspond to high-resolution biological features, such as intron-exon boundaries and transcription factor motifs (preprint: Brixi et al, 2025).

## Benchmark datasets

### Motivation for benchmarking and evaluation efforts

The language models discussed have potential to be useful in genomics and transcriptomics, but questions remain about their performance. These include whether they outperform simple baselines or task-specific methods, and whether a single model can perform well across all tasks.

Most gLM studies perform a thorough analysis of model performance on different datasets when released (Zhou et al, 2024; Dalla-Torre et al, 2025; Fishman et al, 2025, Mendoza-Revilla et al, 2024), comparing against other predictors. However, evaluation tasks, settings, and even results often differ between individual studies, or may not always fully capture the fundamental complexity or sparsity of genomic data (Patel et al, 2024). As a result, a number of independent benchmark datasets have been developed for testing gLMs trained on DNA, RNA, and protein sequences (see Table 3). These datasets enable a standardized and reproducible evaluation of gLMs, similarly to those found in NLP

**Table 3.  Independent benchmark datasets for evaluating genomic language models.**

| Benchmark | Predictive tasks | Models tested | Key conclusions |
|---|---|---|---|
| BEND (Marin et al, 2024) | - Gene finding<br>- Enhancer annotation<br>- Chromatin accessibility<br>- Histone modification<br>- CpG methylation<br>- Noncoding variant effects in disease and expression | NT, DNABERT, DNABERT-2, HyenaDNA, GROVER, GENA-LM, Supervised, Experts | - Linear probing (LP) matched performance by expert methods.<br>- NT and DNABERT achieved best performance overall among gLMs.<br>- Generally poor performance on variant effect prediction and long-range tasks. |
| DART-Eval (Patel et al, 2024) | - Discriminating regulatory DNA sequences<br>- Transcription factor motif sensitivity<br>- Identifying cell-type-specific regulatory DNA<br>- Quantitative prediction of regulatory activity<br>- Predicting the quantitative effect of regulatory genetic variants | Caduceus, DNABERT-2, GENA-LM, HyenaDNA, Mistral-DNA, NT, Supervised | - Fine-tuning systematically gives better performance than LP.<br>- For most tasks, NT, Caduceus and GENA-LM gave the best results.<br>- Supervised methods yield similar results to LP.<br>- gLMs underperformed on variant effect prediction. |
| (Tang et al, 2025) | - Non-coding regulation<br>- Protein-DNA interactions<br>- Zero-shot variant effect<br>- Alternative splicing<br>- RNA pol II elongation<br>- Protein-RNA interactions | NT, GPN, DNABERT-2, HyenaDNA, Experts | - LP underperforms training supervised models, or LP from expert models.<br>- GPN was the only gLM which yielded better results than the baseline. |
| GenBench (Liu et al, 2024) | A collection of 43 short and long-range tasks, covering: non-coding region, coding region, and genome architecture | NT, DNABERT, DNABERT-2, HyenaDNA, Caduceus, GENA-LM | - NT, GENA-LM, and Caduceus perform best on most tasks.<br>- Performance scales with model size, and input length.<br>- K-mer-based approaches are better for gene clustering. |
| TraitGym (preprint: Benegas et al, 2025b) | - Causal non-coding variants for Mendelian traits<br>- Causal non-coding variants for complex traits | GPN-MSA, GPN-Promoter, NT, HyenaDNA, Caduceus, Species-LM, AIDO.DNA, Evo 2, Experts | - gLMs lag behind models with alignment, and GPN-MSA is behind CADD.<br>- Evo 2, Species-LM, GPN-promoter are the best alignment-free models. |
| DNALONGBENCH (preprint: Cheng et al, 2025) | - Predicting enhancer-target gene pairs<br>- eQTL classification<br>- Context map prediction<br>- Recognizing regulatory sequence activity<br>- Transcription initiation prediction | HyenaDNA, Caduceus, Experts, Supervised | - Expert models consistently outperformed gLMs across all tasks, especially on predicting regulatory sequence activity.<br>- gLMs are less efficient at recognizing long-range dependencies. |
| LRB (Kao et al, 2024) | - Variant effect prediction<br>- Gene expression prediction (CAGE, Bulk RNA-Seq) | HyenaDNA, DNABERT-2, DNABERT-S, Caduceus, NT, Experts, Supervised | - FT leads to better performance than LP.<br>- gLMs work reasonably for annotation tasks but underperform on variant effect interpretation and long-range ones. |
| (Boshar et al, 2024) | - Melting point, secondary structure, and stability prediction<br>- Beta-lactamase activity and fluorescence prediction | pLMs, DNABERT-2, NT | - gLMs can match pLMs in some tasks, capturing global patterns in protein sequences.<br>- gLM and pLM representations are complementary to some tasks.<br>- CDS-aware tokenization boosts performance on protein-related tasks. |
| DGEB (West-Roberts et al, 2024) | Several classification, clustering, retrieval and other tasks. 10 tasks as protein sequences, 4 in nucleotides, and 4 in both. | pLMs, NT, Evo | - gLMs representations of coding sequences perform poorly against pLMs.<br>- Model performance does not scale with size for gLMs, only for pLMs. |
| (Zablocki et al, 2025) | 4 RNA secondary structure prediction tasks of increasing complexity and decreasing homology. | RNABERT, ERNIE-RNA, RNAErnie, RNA-FM, RNA-MSM, RiNALMo, Experts, Supervised | - RiNALMo and ERNIE-RNA achieved the best overall performance.<br>- Parameter size, pretraining data size and diversity linked to performance.<br>- On the hardest task, no gLM embedding was more informative than one-hot. |

**Table 3.** (continued)

| Benchmark | Predictive tasks | Models tested | Key conclusions |
|---|---|---|---|
| BEACON (Ren et al, 2024) | - Secondary structure prediction<br>- Contact map prediction<br>- Distance map prediction<br>- Structural score imputation<br>- Splice site prediction<br>- alternative polyadenylation isoform prediction<br>- Non-coding RNA function classification<br>- Modification prediction<br>- Mean ribosome loading<br>- Vaccine degradation prediction<br>- Programmable RNA switches<br>- CRISPR on/off-target prediction | RNABERT, RNA-FM, RNA-MSM, SpliceBERT, UTR-LM, 3UTRBERT (Yang et al, 2024), BEACON-B, Experts, Supervised | - SpliceBERT and RNA-FM achieved best overall results.<br>- gLMs pretrained on relevant RNA attributes perform better.<br>- RNA gLMs benefit from single nucleotide resolution. |
| (You et al, 2025) | - ncRNA classification<br>- N6-methyladenosine prediction<br>- Alternative splicing prediction<br>- Translation efficiency prediction | RNA-FM, RNA-MSM, RNABERT, SpliceBERT, DNABERT, DNABERT-2, RNAErnie, Experts | - gLMs (even DNA-based) perform well against expert methods across all tasks, especially if little fine-tuning data is available.<br>- Model pretraining over parameter size determines performance. RNA-FM for ncRNA tasks; SpliceBERT, where orthologous knowledge is relevant. |
| Genomic Touchstone (preprint: Wang et al, 2025) | 36 tasks focused on human genetics around the central dogma, falling into the following categories:<br>- Genetic function annotation (DNA)<br>- Regulatory mechanism modeling (DNA)<br>- Genetic variant effect prediction (DNA)<br>- Functional studies (RNA)<br>- Post-transcriptional regulation (RNA)<br>- Engineering epplications (RNA)<br>- Structural analysis (Protein)<br>- Functional annotation (Protein)<br>- Property prediction (Protein) | Supervised, NT, DNABERT, DNABERT-2, GPN, HyenaDNA, GROVER, GENA-LM, Caduceus, Mistral-DNA, GENERator, SpliceBERT, RNA-FM, RNAErnie, ESM | - gLMs (NT, GENA-LM, GENERator, Caduceus) achieve the highest performance on most prediction tasks.<br>- gLMs trained on DNA perform well on RNA and protein-based tasks.<br>- Transformer-based models rank highest, but more efficient architectures are competitive with longer sequence contexts.<br>- Design choices influence predictive performance more than parameter counts. |
| mRNABench (preprint: Dalal et al, 2025) | - mRNA half-life prediction<br>- Mean ribosome load prediction<br>- MRL-HL pair prediction<br>- GO term classification prediction<br>- mRNA localization prediction (LR & SR)<br>- Protein localization prediction<br>- MRL-MPRA prediction<br>- eCLIP binding prediction<br>- Variant effect prediction | 3UTRBERT, AIDO.RNA, DNABERT-S, DNABERT-2, ERNIE-RNA, Evo, Evo 2, HelixmRNA, HyenaDNA, NT, Orthrus, RiNALMo, RNABERT, RNAErnie, RNA-FM, RNA-MSM, SpliceBERT, UTR-LM, Supervised | - Small Mamba-based models outperform larger ones, showing architecture and objectives matter more than scale.<br>- mRNA-trained models outperform DNA/ncRNA ones, revealing key distributional differences.<br>- Naive splits overestimate generalization; biologically aware splits reveal current model limitations. |

benchmarks (Wang et al, 2019; Rajpurkar et al, 2016). Independent benchmarking efforts are listed in Table 3. For brevity, datasets released with gLMs (Zhou et al, 2024; Dalla-Torre et al, 2025; Fishman et al, 2025, Mendoza-Revilla et al, 2024) are not shown in the table, nor are benchmarks without any tested gLMs, i.e., GuANINe (Robson and Ioannidis 2023) and Genomic Benchmarks (Grešová et al, 2023).

### Benchmark datasets for RNA language models

For RNA modeling, notable benchmarks include BEACON (BEnchmArk for COmprehensive RNA Task and Language Models) (Ren et al, 2024), encompassing 13 diverse tasks across structural analysis, functional studies, and engineering applications. Evaluations via fine-tuning on BEACON show that pretrained RNA language models (mainly RNA-FM (preprint: Chen et al, 2022) and SpliceBERT) outperformed previous task-specific state-of-the-art

methods on 8 out of the 13 tasks (see Table 3), although supervised (ResNet and LSTM) baselines also work well. A complementary benchmark focusing on RNA secondary structure prediction (Zablocki et al, 2025) found that embeddings from larger models such as RiNALMo and ERNIE-RNA generally lead on structure prediction. However, it also revealed a large performance gap between current gLMs and classical, homology-based methods, especially in challenging, low-homology scenarios where gLMs may even underperform a simple one-hot encoding. Such findings were supported by a benchmark (You et al, 2025) that tested gLMs understanding of RNA properties, covering processes such as ncRNA classification, m6A modification prediction, alternative splicing prediction, and translation efficiency prediction, in different prediction settings (sequence-level, nucleotide-level, regression). This study highlights that larger gLM models do not necessarily fare better than specialized models with relevant

pretraining strategies and learning objectives. For example, RNA-FM (trained on RNAcentral) performs best on ncRNA classification, SpliceBERT performs well on m6A modification prediction and splicing tasks (leveraging evolutionary conservation). The gap between gLMs and other methods was more pronounced with limited data available, and interestingly, even DNA-based gLMs could predict RNA properties with high accuracy (Ren et al, 2024; preprint: Wang et al, 2025).

### Benchmark datasets for DNA language models

In contrast to RNA benchmarks, evaluations of DNA gLMs paint a more mixed picture, highlighting limitations in generalization and consistent task performance. A number of benchmarks (Marin et al, 2024; Liu et al, 2024; Dalla-Torre et al, 2025; Zhou et al, 2024; Ren et al, 2024; preprint: Feng et al, 2024; West-Roberts et al, 2024a) have been presented to test the general predictive power of gLMs on DNA sequences, containing multi-purpose datasets. For example, the Benchmark for DNA Language Models (BEND) (Marin et al, 2024) focuses on the human genome, including tasks like gene finding, chromatin accessibility, and variant effect prediction. Downstream performance obtained via linear probing (BEND) and fine-tuning (Zhou et al, 2024, Patel et al, 2024) suggest that there is no single all-purpose gLM; even so, the NT (Multispecies), GENA-LM, and GENERator (preprint: Wu et al, 2025), are strong default choices for most short-range prediction tasks on DNA. The SSMs, such as Caduceus and HyenaDNA, underperform larger models, yet if input sequences are not length-constrained, larger context sizes can narrow the gap between SSMs and transformer-based models (preprint: Wang et al, 2025).

DNA-based gLMs have also been tested on RNA (Ren et al, 2024, preprint: Wang et al, 2025), and protein (Boshar et al, 2024; Marin et al, 2024) prediction tasks, with promising results compared to domain-specific gLMs. Other studies (Kao et al, 2024; preprint: Cheng et al, 2025) suggest that all gLMs struggle to model distal interactions on DNA (such as enhancer-gene pairings, and the effects of eQTLs), often underperforming sequence-to-function models. In fact, most studies find that although gLMs tend to be more accurate than simple supervised baselines, in tasks where a sophisticated, specialized predictive model is available, gLMs do not consistently meet its performance, or only outperform them by small margins (preprint: Cheng et al, 2025; Boshar et al, 2024; Marin et al, 2024).

## Challenges and future developments

### Empirical limitations of current genomic language models

It is important to determine whether the performance gains stem from the improved architectures or from the pretraining process. One study reported that seeding gLMs with the pretrained weights when fine-tuning on several benchmarks did not consistently lead to better performance than from random starting values (preprint: Vishniakov et al, 2024), questioning the utility of learned representations through language modeling. Alternatively, this may be explained by gLMs undergoing catastrophic forgetting (Kirkpatrick et al, 2017), whereby their pretrained knowledge is overwritten by the latest training phase if not fine-tuned correctly (e.g., using a too large learning rate). This phenomenon has also been observed in NLP and is an active research area (Luo et al, 2025b; preprint Süalp and Rezaei, 2025).

Capturing long-range genomic interactions presents a further challenge (Marin et al, 2024; preprint: Vishniakov et al, 2024), as gLMs struggle to match the performance of specialized expert methods on tasks requiring extensive contextual understanding, such as enhancer annotation and 3D genome organization prediction (preprint: Cheng et al, 2025; Kao et al, 2024). This limitation is exacerbated by the quadratic computational costs of transformer-based approaches and the extreme sparsity and length scales of genomic signals in many eukaryotic genomes (Marin et al, 2024). Additionally, gLMs demonstrate notable deficiencies in recognizing cell-type-specific regulatory signals, showing poor cell-type separation in zero-shot embeddings compared to simpler motif-based baselines and basic supervised CNNs (Patel et al, 2024).

Several benchmarks (Dalla-Torre et al, 2025; preprint: Benegas et al, 2025b; Liu et al, 2024) further reveal that many pretrained gLMs show limited sensitivity to genetic mutations, often producing nearly identical embeddings for heavily mutated sequences, which undermines their effectiveness in mutation-dependent tasks such as pathogenicity prediction. Models trained with evolutionary constraints, such as the GPN family, show stronger performance, particularly for Mendelian traits (preprint: Benegas et al, 2025b), but even these struggle to consistently outperform traditional tools like CADD (Combined Annotation Dependent Depletion) (Rentzsch et al, 2019) and PhyloP (Pollard et al, 2010), two robust baselines in identifying deleterious variants, based on functional and population genomics (CADD), and evolutionary conservation (PhyloP). Notably, pLMs show greater sensitivity to missense mutations, as demonstrated by the high performance on zero-shot variant effect prediction tasks (Notin et al, 2023). PLMs likely infer phenotype from structural effects (Zhang et al, 2024c), which suggests that variant effects in non-coding DNA should also be estimated via their effects on function.

The interpretability challenges facing gLMs are equally pressing (Sarumi and Heider, 2024), as their black-box nature makes it difficult to distinguish between genuine genomic understanding and statistical memorization of training patterns (Consens et al, 2025b; Sanabria et al, 2024b). Attention score visualization, the primary interpretability method for genomic transformers, proves inadequate for explaining the full complexity of model computations (Consens et al, 2025b), while newer architectures like SSMs lack established interpretability frameworks entirely.

### Experimental confirmability and evaluation bias

Many reported gains for gLMs remain confined to computational benchmarks, with few studies prospectively testing model-driven hypotheses in wet-lab assays. Positive examples demonstrate feasibility: Evo-generated sequences were shown to yield functional CRISPR-Cas molecular complexes as well as IS200/IS605 transposable systems (Nguyen et al, 2024), and UTR-LM prospectively validated predictions by testing a library of 211 engineered 5′ UTRs with high predicted translation in reporter assays (Chu et al, 2024). A cost-efficient alternative to this would be the in silico assessment of model predictions via established state-of-the-art specialized models, for example, verifying structural plausibility via AlphaFold 3 (Abramson et al, 2024). Evaluation efforts of gLM studies rely on existing computational datasets and normally do not incorporate tasks that directly translate to outcomes of wet-lab experiments. Benchmarks using deep mutational scan, mutagenesis, or reporter

data (Notin et al, 2023; Tang et al, 2025; (Robson and Ioannidis 2023) bring evaluations closer to DNA/RNA function than purely in silico metrics, and should lead the field towards more applicable models.

We note that most evaluation efforts tend to focus on the same set of gLMs, largely overlooking others, e.g., GeneSLM (Zvyagin et al, 2022), MegaDNA (Shao and Yan, 2024), LucaOne (He et al, 2025), DeepGene (Zhang et al, 2024a), which may have different approaches to genome modeling. Moreover, commonly used pretraining and benchmark datasets exhibit nontrivial biases (e.g., taxonomic and assay over-representation, homologous sequences), which can inflate apparent performance and limit out-of-distribution generalization. Nevertheless, these results question the current approach to genome foundation models. As others (Patel et al, 2024), we suggest that there are certain requirements that gLMs should meet in order to be of use for genomic research, and which should be the target of comprehensive evaluation efforts. While specific details may vary based on the intended application domain of a gLM, general requirements are summarized in Box 2.

### Developing next-generation gLMs

These persistent limitations largely reflect a mismatch between current gLM assumptions and the structure of genomic information. Functional signals are sparse and highly context-dependent, yet reconstruction-style pretraining on unlabeled whole genomes treats all bases as equally informative. This encourages models to memorize local redundancy in repetitive DNA and overlook rare, cell-type-specific regulatory cues and single-nucleotide effects. Addressing this will likely require: (i) stronger biological inductive biases in pretraining (e.g., region-aware losses that prioritize functional bases such as cis-regulatory elements; homology/evolution-informed objectives); (ii) mutational risk-aware tasks to improve single-base resolution modeling; (iii) hybrid or modular architectures that couple efficient long-range modeling with priors from regulatory genomics and/or

---

**Box 2    Minimal requirements for genomic language models**

Biologically grounded representations from pretraining
  *Pretraining tasks and data should be selected in order to maximize teaching the model semantically rich and biologically meaningful DNA/RNA representations that are relevant for downstream applications, forming the basis for transfer learning.*

  Efficient and flexible transfer learning
  *The gLM should be designed to transfer easily to new tasks, with minimal architectural changes or hyperparameter tuning. It must remain effective when applied on small datasets, common in many biological domains. Efficient adaptation ensures practical utility across a wide range of applications.*

  Pretraining advantage over random initialization
  *A pretrained gLM should consistently outperform the same model architecture trained from scratch. This confirms meaningful advantages of embedding prior genomic knowledge into the model parameters, thus justifying the effort and computational cost of pretraining.*

  Superiority to traditional supervised baselines
  *The gLM should achieve on par or higher accuracy than traditional supervised methods, trained on the same dataset. This demonstrates that the learned representations of DNA/RNA are more informative than shallow features and reinforces the value of pretraining.*

---

structure; and (iv) multimodal integration (e.g., genomic DNA and histone modification signals) to supply missing domain context, particularly with regard to local chromatin state.

The emergence of large-scale pangenome graphs offers an opportunity to expose models to population-level variation, structural variants, and haplotype context that have been largely absent from current training regimes. Unlocking these benefits will depend on advances in graph-aware representations, phasing-informed tokenization, and evaluation frameworks that account for redundancy and relatedness. We anticipate that such methods will enable gLMs to learn efficiently from pangenome-scale diversity.

## Applications in genomic research

This section explores the diverse applications of gLMs in genomic research, highlighting their potential through generation and transfer learning. These models are being used for tasks ranging from genomic sequence annotation and regulatory modeling to variant effect prediction. Additionally, gLMs are contributing to RNA structure prediction and advancing the understanding of genomic diversity in areas like metagenomics.

### Biological sequence generation and design

One promising application of gLMs is their use for de novo sequence generation tasks, which can be done via recursively predicting the next token in the sequence, starting from a control token that guides the design, or by providing an incomplete sequence that needs to be completed (Fig. 3A). Indeed, some models have already demonstrated capabilities in designing cis-regulatory elements (CREs). For example, both AIDO.DNA and regLM learned part of the regulatory grammar in yeast, allowing for the directed generation of sequences with tunable transcriptional activity (Ellington et al, 2024; Lal et al, 2024). RegLM, along with GENERator, can also be used for designing human enhancers with desired activity, specific to particular cell lines (K562, HepG2, and SK-N-SH) with regLM (preprint: Wu et al, 2025). The viability of these generated CREs can be assessed via enrichment for cell-type-specific transcription factor motifs, and with sequence-to-function models trained to predict CRE activity.

GLMs have also shown strength in coding sequence generation and the directed design of sequences with targeted molecular or epigenomic properties. The GENERator model can accurately generate protein-coding DNA, yielding sequences that follow natural codon structure and encode proteins homologous to known families like histones or cytochrome P450s (preprint: Wu et al, 2025). Evo 2 can also perform a gene completion task, whereby it can generate the remaining sequence of a gene when prompted with both upstream context and an initial portion of a highly conserved protein (preprint: Brixi et al, 2025). Moreover, Evo 2 can be coupled with sequence-to-function models to yield DNA with desired properties, such as controlling for the location and length of accessible chromatin within it.

In the case of RNA sequences, GenerRNA was specifically designed for RNA generation. It has been pretrained on 11.6 billion nucleotides and can be used to generate novel RNA sequences with stable secondary structure. With fine-tuning, it can create RNA

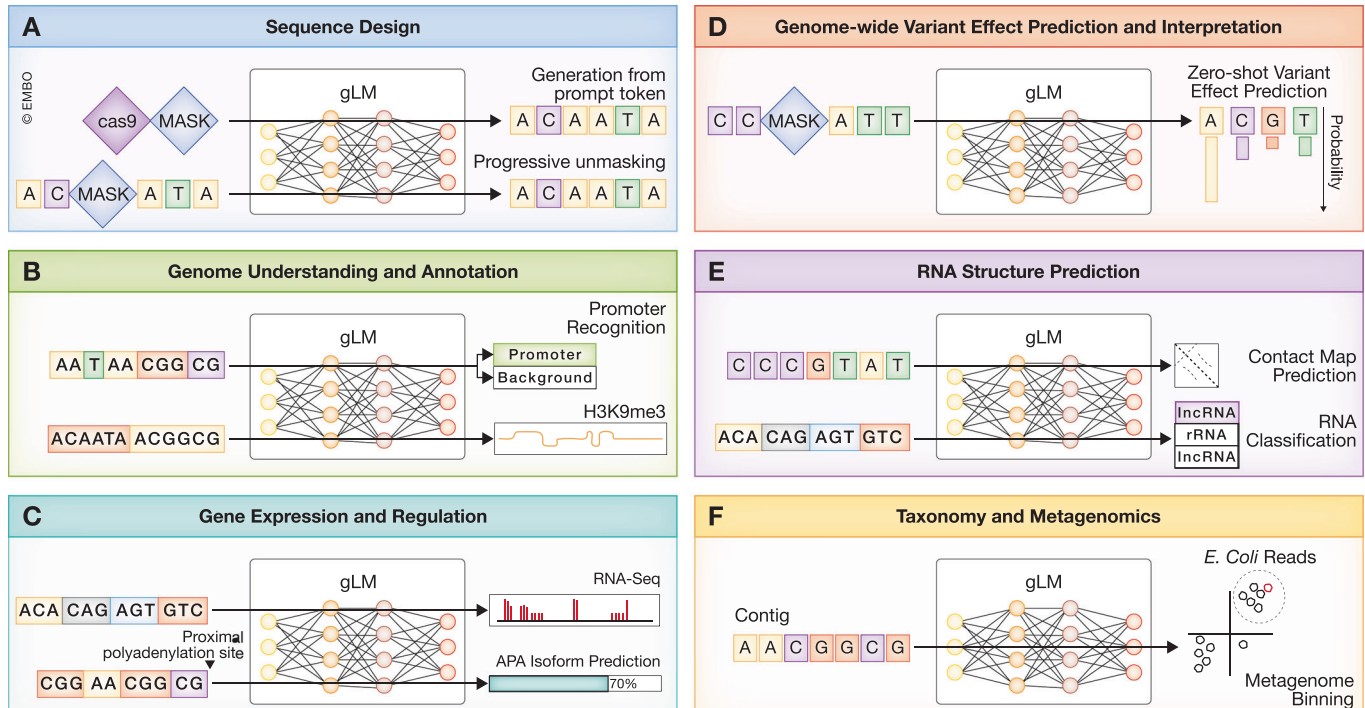

**Figure 3. Application areas of genomic language models.**

(A) New sequences can be generated from a control token (e.g., *Cas9* complex) by iteratively predicting the next token or completed from partial input via progressive unmasking. (B) Learned representations from gLMs enable annotation of genomic regions, such as distinguishing promoters from background or predicting epigenomic features like H3K9me3 status. (C) Gene regulatory tasks can be modeled by fine-tuning gLMs to predict outputs like bulk RNA-seq signal or alternative polyadenylation (APA) isoform usage from sequence. (D) As zero-shot variant effect predictors, gLMs estimate the impact of mutations using the log-likelihood ratio between reference and alternate alleles. (E) RNA structure can be inferred using gLMs to predict contact maps and classify structural families, revealing long-range and functional interactions. (F) In metagenomic binning, gLMs support taxonomic classification by assigning sequence reads to species; here, a read is correctly identified as originating from *E. coli*.

sequences with tunable binding affinity to proteins like ELAVL1 and SRSF1 (Zhao et al, 2024). Evo supports co-generation of RNA-protein complexes, including functional Cas9 variants, by leveraging token-conditioned generation on thousands of CRISPR-Cas loci (Nguyen et al, 2024).

More broadly, some gLMs (Nguyen et al, 2024; preprint: Brixi et al, 2025; Shao and Yan, 2024; preprint: Touvron et al, 2023) demonstrated the capacity to generate DNA sequences with realistic features across different species. MegaDNA, trained on unannotated bacteriophage genomes, can produce de novo sequences up to 96 kb long that contain biologically plausible gene densities (Shao and Yan, 2024). In bacteria, Evo and Evo 2 can generate megabase-scale sequences that reproduce natural genome features such as operon-like organization, GC content, codon usage, and complete tRNA sets (Nguyen et al, 2024; preprint: Brixi et al, 2025). Evo 2 (and GENERator) are also applicable to eukaryotes, producing yeast chromosomes and mitochondrial genomes with accurate intron-containing genes, promoter positioning, and gene synteny (preprint: Brixi et al, 2025; preprint: Wu et al, 2025). Additionally, DNAGPT has been tested on de novo generation of artificial human genome segments by fine-tuning on haplotype data, producing sequences that replicate allele frequencies and linkage disequilibrium found in real genomes (preprint: Touvron et al, 2023).

## Genomic sequence understanding and annotation

GLMs have also demonstrated promising capabilities at recognizing diverse genomic elements. Results on benchmark datasets (Patel et al, 2024; Zhou et al, 2024) suggest that gLMs can distinguish regulatory DNA (promoters or enhancers) from synthetic background sequences (Fig. 3B). Amongst the evaluated models, NT (multispecies) and DNABERT-2 showed particularly strong performance when fine-tuned, whereas GENA-LM and Caduceus excelled more in zero-shot settings (preprint: Wang et al, 2025; Patel et al, 2024; Marin et al, 2024). Performance via probing matched the previous supervised approaches. NT, GENA-LM, and GENERator, demonstrate strong performance in identifying promoters (preprint: Wang et al, 2025, Liu et al, 2024), their types, and core promoter regions. Similarly, NT and DNABERT-2 recognize frequent transcription factor binding sites in the human and mouse genomes in zero-shot settings (Patel et al, 2024) and can successfully identify human enhancer sequences (Marin et al, 2024), but with performance similar to tested baselines (preprint: Wang et al, 2025). These findings suggest that gLMs can be readily applied for annotating regulatory DNA, either fine-tuned, if labeled data is available, or zero-shot.

Performance on epigenomic prediction tasks is heavily dependent on the epigenomic feature predicted, and if the model is fine-

tuned. DNA methylation prediction represents the most successful application area, with nearly all gLMs achieving exceptional performance on CpG methylation tasks (preprint: Wang et al, 2025), perhaps expected given lower DNA methylation at CpG islands, elements with strong sequence-encoded signals. For chromatin accessibility prediction, NT has demonstrated strong performance in predicting quantitative tracks (such as ATAC-seq and DNase-seq) across cell types compared to other gLMs and supervised baselines (Patel et al, 2024; preprint: Wang et al, 2025). Histone modification prediction can be the more challenging epigenomic task, with models (Dalla-Torre et al, 2025; Zhou et al, 2024a) achieving varying performance across different histone marks (Zhou et al, 2024a; Dalla-Torre et al, 2025; preprint: Wu et al, 2025).

SegmentNT (de Almeida et al, 2025a, 2025b) has also demonstrated how gLMs like NT can be leveraged for fine-grained genome annotation and surpass traditional tools such as AUGUSTUS (Stanke et al, 2008) in gene finding, particularly when segmenting whole genomes or capturing all gene isoforms. It also excelled at splice site recognition, a task where the base NT model has also achieved high accuracy, challenging the task-specific model SpliceBERT (preprint: Wu et al, 2025). For non-coding RNA classification, models such as Caduceus and RNA-FM lead performance benchmarks due to targeted pretraining on ncRNA data, with SpliceBERT offering an efficient alternative with strong performance (preprint: Wu et al, 2025; You et al, 2025).

## Gene expression and regulation

Gene RNA expression prediction (Fig. 3C) is one of the most challenging gLM applications, as it requires an accurate understanding of both regulatory DNA and epigenomic signals. In bulk RNA expression prediction, some gLMs (Dalla-Torre et al, 2025, Fishman et al, 2025) approach the performance of sequence-to-function models like Enformer (Liu et al, 2024, Kao et al, 2024), suggesting that pretrained language models can capture broad expression patterns from sequence alone. A large context size is helpful for this task, and models improve performance when provided with longer input sequences, with architectures like HyenaDNA demonstrating particular strength by leveraging extended genomic context spanning up to 100 kilobases (Kao et al, 2024). For enhancer-promoter interaction prediction, gLMs perform strongly, with DNABERT-2 and Caduceus variants achieving high classification accuracies, indicating the models' potential to generalize across complex regulatory interactions (preprint: Wang et al, 2025). In transcript and protein abundance prediction, GENERator and NT lead performance, again demonstrating that gLMs can approximate or even match expert-designed models in certain settings (preprint: Wang et al, 2025).

In related tasks, such as mRNA stability and RNA modification prediction (Fig. 3C), several gLMs also perform well, particularly Orthrus, GENERator and SpliceBERT, the latter excelling at m6A modification prediction (preprint: Wang et al, 2025; You et al, 2025, preprint: Dalal et al, 2025). The benefit of gLMs becomes more apparent when only limited labeled data were available, for example, Orthrus showcased superior generalization in mRNA half-life and translation efficiency, including from just a hundred training samples (Fradkin et al, 2024). Some DNA-based models

also demonstrated understanding of RNA regulation, for example, Caduceus models outperformed several supervised baselines, while maintaining competitive results with state-of-the-art expert methods (preprint: Wang et al, 2025) in alternative polyadenylation isoform task. However, in some cases, these task-specific supervised methods still hold a substantial edge. For instance, in transcription initiation prediction, such as CAGE profiling, gLMs lag significantly behind the supervised, sequence-to-function models (preprint: Cheng et al, 2025; Kao et al, 2024).

Performance gaps also remain in enhancer activity prediction, where gLMs show limited gains over standard sequence-based approaches and underperform in comparison to specialized models, especially when tasked with cell-type-specific regulatory activity prediction (preprint: Cheng et al, 2025; preprint: Wang et al, 2025; Patel et al, 2024). This underscores the need for further pretraining innovations and, possibly, multimodal integration to bridge the gap between gLMs and task-specific supervised methods.

## Genome-wide variant effect prediction and interpretation

Genome-wide variant effect prediction (Fig. 3D) is essential for identifying causal genetic variants that influence gene regulation, disease risk, and trait heritability. While gLMs have taken steps in gene expression modeling, they generally fall short in noncoding variant effect prediction compared to functional genomics-supervised methods (preprint: Cheng et al, 2025; Kao et al, 2024). In tasks like expression QTL and chromatin QTL prediction, models such as Enformer, Borzoi, and ChromBPNet consistently outperform gLMs (Manzo et al, 2025; preprint: Cheng et al, 2025), benefiting from functional genomics training or architecture optimized for fine-grained regulatory features. Although a few gLMs (preprint: Wu et al, 2025; Schiff et al, 2024) show competitive performance in coding and noncoding variant prediction, overall sensitivity to subtle sequence changes remains a limitation.

In disease variant prediction tasks, frameworks like DYNA (Zhan et al, 2025) apply targeted fine-tuning, outperforming general-purpose genomic language models on both coding and noncoding variants. This results from leveraging pretrained representations from pLM and gLMs, followed by additional training to distinguish between tolerated and disease-associated sequence variations. For Mendelian traits, the recent alignment-based GPN-MSA model, and older conservation-aware models such as CADD, or simply the conservation score phyloP continue to set the standard (preprint: Benegas et al, 2025b). While the newer, large-scale Evo 2 shows improvement among alignment-free approaches (preprint: Benegas et al, 2025b), gLMs currently lag behind in capturing the precise, functional impact of genomic variants, highlighting the opportunity for improvement via multimodal training and enriched representation of evolutionary and regulatory context.

## RNA structure prediction

A large fraction of RNA sequences does not code for protein products. Known as noncoding RNAs (ncRNAs), these sequences adopt specific structures to perform important biological functions. A long-established challenge of computational genomics is to determine the structures of these ncRNAs. Similar to the observed

trends in pLMs for protein folding, RNA-based gLMs are also greatly applicable to structure prediction tasks.

In secondary structure prediction (Fig. 3E), the goal is to identify base pairs that form the RNA secondary structure, such as stems and loops. Models like RiNALMo (Penić et al, 2025) and ERNIE-RNA (Yin et al, 2025) have consistently ranked among the top RNA-specific gLMs, particularly on benchmarks with moderate sequence similarity, demonstrating competitive scores even against expert methods (Zablocki et al, 2025). However, on more challenging benchmarks involving novel RNA families or real-world PDB-derived sequences, gLMs often underperform compared to classical folding methods, indicating limitations in generalization to RNAs distant from those with known structures (Zablocki et al, 2025). In contact map prediction, models try to identify which nucleotide pairs are spatially proximate in the RNA's 3D structure. ERNIE-RNA, in particular, shows superior performance due to its modified attention maps that incorporate structural base-pairing information on top of the raw sequence into the computation, especially for long-range contacts (Yin et al, 2025). In addition, a recent model RNAGenesis surpassed the state-of-the-art at the BEACON contact map prediction task (among others) using fine-tuning with LoRA (preprint: Zhang et al, 2025).

RNAGenesis, along with AIDO.RNA also shows superior performance at a related task, identifying families of RNA structures (Fig. 3E), revealing families of structures (preprint: Zhang et al, 2025; Zou et al, 2024; Ren et al, 2024). Interestingly, even gLMs, like Caduceus, show strong clustering performance on RNAs (preprint: Wang et al, 2025). These models demonstrate that with appropriate pretraining strategies, gLMs can encode sequence features necessary for downstream inference of structure, often matching or exceeding specialized models.

### Taxonomy and metagenomics

Further use cases for gLMs include taxonomic classification and metagenomics applications, with several specialized architectures developed for microbial community analysis (preprint: Zhou et al, 2025a; Liu et al, 2025b; preprint: Zhang et al, 2025; Gwak and Rho 2022; Zvyagin et al, 2022). Some more general models like HyenaDNA excel at species classification by leveraging extended sequence contexts (Nguyen et al, 2023), while GenomeOcean and DNABERT-S show particular strength in metagenomics binning tasks (Fig. 3F), effectively clustering and discriminating between different microbial species from DNA sequences alone (preprint: Zhou et al, 2025a; Zhou et al, 2025b).

For specialized public health applications, METAGENE-1 is designed as a metagenomic foundation model for pandemic monitoring and pathogen detection, learning from the full genomic distribution present in environmental samples such as wastewater (Liu et al, 2025b). Similarly, ViBE demonstrates robust performance in viral classification, even for previously unknown viruses, by learning comprehensive viral genomic context directly from metagenome sequencing data (Gwak and Rho 2022). Lastly, GenSLMs is designed to learn the evolutionary landscape of whole genomes, particularly for identifying and classifying new and emergent variants of pandemic-causing viruses like SARS-CoV-2 (Zvyagin et al, 2022). These specialized applications highlight the versatility of gLMs in addressing complex challenges in microbial

ecology and public health surveillance, where rapid and accurate identification of organisms from mixed environmental samples is crucial.

## Conclusions and open questions

We have seen the emergence of a large number of gLMs in recent years, with different architectures and design choices. These models can be powerful tools for many applications, yet most models currently do not meet the promise of an out-of-the-box "foundation model" suitable for all tasks. Indeed, a relevant pretraining scheme is the biggest influence on downstream performance for a specific task. Consequently, there seems to be a trend towards more specialized models explicitly designed for certain biological applications. To aid practitioners, we provide a short guide to selecting the right model presented in Box 3.

It is an open question whether the foundation model paradigm underlying gLMs will see a revival with more advanced training strategies or architectures. An important obstacle in modeling the elusive "grammar" of eukaryotic genomes is that the DNA sequence in many loci has only a weak functional effect and is thus free to vary. However, it is nonetheless non-random, where patterns instead stem from a neutral background of mutational processes; these are a strong distraction to language model training, diverting their attention from comparatively rarer regulatory elements. In addition to the challenge of adequately factoring out neutral mutagenesis, another challenge is effectively incorporating inter-individual variation during gLM training. While it is generally recognized that learning from multiple individuals, e.g. the 1000 Genomes Project, is beneficial, the best way to integrate this data is unclear. Moreover, since human population variation is modest, it is of interest how best to draw on evolutionary data. Multiple sequence alignments were shown to be highly informative for gLMs but are difficult to set up for more rapidly evolving sequences, including the arguably more interesting regulatory DNA. New approaches to combine data across evolutionary distances may significantly boost pretraining.

GLMs also hold substantial promise for clinical applications (Consens et al, 2025b), via decoding the genomic regulatory mechanisms to advance our understanding of human disease and transform personalized care. For example, predicting the effects of non-

---

**Box 3    Practitioner's short guide to selecting the right gLM**

**Variant effect prediction**. *Prioritize models trained with single-nucleotide tokens, and with evolutionarily-informed objectives (e.g. GPN family, Orthrus, NT-multispecies). Avoid BPE-based models where the variant might be merged into a larger token.*

**Gene/genomic element annotation**. *DNABERT-2 and GROVER, based on BPE tokenization, remain robust generalist choices for identifying promoters and functional elements in short-to-medium contexts.*

**Long-range tasks (~>10 kb)**. *Select SSM-based (HyenaDNA, Caduceus) or hybrid (e.g., Evo 2) architectures. Transformer-based models may become computationally prohibitive and show diminishing returns at this scale.*

**Limited compute and/or large datasets**. *Use domain-specialized models (e.g., Species-LM for fungi, Orthrus for RNA) which often outperform larger generalist models on their specific tasks while requiring a fraction of the inference cost.*

coding variants via gLMs could be integrated with existing genomic testing pipelines to flag pathogenic regulatory variants that current screening methods might miss. In the foreseeable future, however, gLM predictions would need to be validated by functional assays before meeting standards for clinical use. An application that bears mentioning is the prediction of effects of GWAS hits, particularly in light of identifying affected enhancers and their downstream target genes, which can elucidate disease mechanisms, improve polygenic risk scoring, and prioritize new drug targets. Additionally, the generation capabilities of gLMs hold potential to accelerate the development of new gene therapies through the computational design of DNA sequences with desired biological functions.

Although they hold great promise, gLMs still face significant technical hurdles before widespread adoption as foundation models or for more specific applications. Many current evaluation approaches rely on simple benchmarks that underestimate the complexity of genomic regulation (to some extent also because many mechanisms are unknown), or do not accurately distinguish true "understanding" (which would be demonstrated by generating unnatural but functional sequences) from memorization of existing sequence classes. Benchmarks also do not offer an established way of testing successful integration of genomes from multiple species or individuals of a population, so as to assess the understanding of their underlying evolutionary logic. Many models demonstrate an insensitivity to single-nucleotide changes or mutations, often yielding high embedding similarities even when substantial portions of the DNA sequence are altered (preprint: Vishniakov et al, 2024). Ethical concerns are important, encompassing issues such as privacy and informed consent in whole-genome variant screening for inclusion in gLM training data, the dual-use potential of research tools being misused for harmful purposes, and inequities in access to gLMs stemming from the potentially high computational demands of future models.

The transformative potential of gLMs in biological and medical research remains evident. Realizing this potential, however, necessitates continued efforts in both developing more accurate, genetic variation-aware, and interpretable models and the establishment of robust biologically informed evaluation standards for their use in real-world settings.

# Peer review information

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

## Acknowledgements

MV received the support of a fellowship from the "la Caixa" Foundation (ID 100010434), with fellowship code (LCF/BQ/DR23/12000017). We thank Luka Velimirov and Noah Wolford for their insightful comments on the manuscript. LLM-assisted tools were used for note-taking and for language editing of manuscript text. Work in the lab of FS is supported by an ERC Consolidator Grant "STRUCTOMATIC" (101088342), Horizon2020 RIA project "DECIDER" (965193), Horizon Europe project "LUCIA" (101096473), CaixaResearch project "POTENT-IMMUNO" (HR22-00402), a Novo Nordisk Fonden "Start Package" grant, the Danish Cancer Society grant "AI-DRIVERS" and a DFF Project2 (5243-00072B).

## Author contributions

**Marcell Veiner**: Conceptualization; Investigation; Writing—original draft; Writing—review and editing. **Fran Supek**: Conceptualization; Supervision; Investigation; Writing—original draft; Writing—review and editing.

## Disclosure and competing interests statement

The authors declare no competing interests.

