## [Peer Review File · Molecular Systems Biology]

The DNA Dialect: A Comprehensive Guide to Pretrained Genomic Language Models

Marcell Veiner and Fran Supek

Corresponding author(s): Fran Supek (fran.supek@bric.ku.dk)

Review Timeline:

Submission Date:	22nd Aug 25
Editorial Decision:	30th Oct 25
Revision Received:	28th Nov 25
Accepted:	2nd Dec 25

Editor: Jingyi Hou

Transaction Report:

30th Oct 2025

Manuscript Number: MSB-2025-13304

Title: The DNA Dialect: A Comprehensive Guide to Pretrained Genomic Language Models

Author: Marcell Veiner

Fran Supek

Dear Fran,

Thank you for submitting your Review article to Molecular Systems Biology. We have now received reports from two reviewers who agreed to evaluate your manuscript.

Both reviewers found your manuscript to be comprehensive, timely, and very well-written. However, they also raised several points and suggestions that could help further strengthen the paper. We strongly encourage you to carefully consider all the feedback during your revision, while leaving it to your discretion whether or not to incorporate specific suggestions.

On a more editorial level:

1. Please submit the manuscript in .docx format. Remove all figures from the manuscript file and upload each as a separate, high-resolution image file. Our graphic designer will redraw the figures based on these.
2. Provide up to five keywords in the manuscript file.
3. Place the figure legends below the References.
4. Include a "Disclosure and Competing Interests Statement." Any employment in a biotechnology company must be explicitly stated in this section.
5. Add funding information in the Acknowledgements section, ensuring it is consistent with the details entered in the online submission system.
6. Supplementary Tables:
 - Update the source file name, title, legend, and manuscript callout to Dataset EV1 (previously Supplementary Table 1). The legend should be included as a separate tab/sheet within the same Excel file.
 - Rename Supplementary Table 2 to Table EV1. The legend should be uploaded as a separate tab/sheet in the Excel file.
7. Please do not upload the Excel files for Tables 1-2 in your next submission. These tables should be included only within the manuscript file.
8. Add the missing callouts for Fig. 2D, Fig. 3C-F, and Table EV1 (formerly Supplementary Table 2).
9. Section order should be arranged in the following order: Title page - Abstract -Keywords - Introduction - Acknowledgements - Disclosure and Competing Interests Statement - References -Figure Legends - Table(s)

When submitting a revised version of your manuscript, please attach a covering letter giving details of the way in which you have handled each of the points raised by the referees. A revised manuscript will be once again subject to review.

We look forward to receiving the revised manuscript soon.

Kind regards,
Jingyi

Jingyi Hou, PhD
Senior Editor
Molecular Systems Biology

We realize that it is difficult to revise to a specific deadline. In the interest of protecting the conceptual advance provided by the work, we recommend a revision within 3 months (28th Jan 2026). Please discuss the revision progress ahead of this time with

the editor if you require more time to complete the revisions. Use the link below to submit your revision:

*** PLEASE NOTE *** As part of the EMBO Press transparent editorial process initiative (see our Editorial at <https://dx.doi.org/10.1038/msb.2010.72>), Molecular Systems Biology publishes online a Review Process File with each accepted manuscripts. This file will be published in conjunction with your paper and will include the anonymous referee reports, your point-by-point response and all pertinent correspondence relating to the manuscript. If you do NOT want this File to be published, please inform the editorial office at contact@molsystbiol.org within 14 days upon receipt of the present letter.

Reviewer #2:

Veiner and Supek provide a comprehensive overview of existing genomic language models (gLMs) and compare them across multiple aspects. I found the manuscript to be well-written, detailed, and mostly correct (see below). I have just a few comments:

1. Paragraph 499-505 is confusing. In line 499, the authors say "Encoder-only transformers mostly rely on the GPT", but GPT is a decoder-only model, so I assume that they wanted to write "Decoder-only transformers". However, in lines 504-505 they mention DNABERT2 and RiNALMo, which are encoder-only. If the reason for mentioning DNABERT2 and RiNALMo is their usage of FlashAttention, I suggest this be rephrased (or moved elsewhere), so that it is not assumed that they are also decoder-only.

2. While Tables 1 and 2 do a good job at presenting the comparison systematically, parts of this manuscript are text-heavy and somewhat difficult to read. The authors may consider creating one such (smaller) table for each section, e.g., Learning Objective, Architecture, etc. In my opinion, these tables would serve as summaries of each section and would help readers to better keep track of how different models operate

3. gLMs are presented mostly as computational achievements and there is a lack of attention to experimental validation. For example, it is stated that RiNALMo outperforms other RNA foundation models, but the review doesn't address experimental confirmability, dataset biases, or whether predictions translate into functional assays.

4. The review lists several limitations of existing gLMs, (e.g., mutation insensitivity and long-range interaction). It would be interesting to see the analysis of why these failures persist and what conceptual innovations are required to address them.

5. Related to the previous point, the "Criticisms" section identifies issues such as catastrophic forgetting or weak mutation sensitivity, but does not weigh how critical they are relative to other fields (e.g., protein language models and NLP).

6. I would be interested to see if the authors could more explicitly give their opinion on how the field of gLMs will move forward. For example, will the future best-performing models also be mainly Transformers, or do you think it will hybrid architectures reach practical use? How will pangenome efforts reshape gLM training datasets.

7. Typo in line 161: orgnaisms -> organisms

8. Line 166: GD -> GLM? If not, then where is the abbreviation GD defined?

Reviewer #3:

Review of "The DNA Dialect: A Comprehensive Guide to Pretrained Genomic Language Models"

Overview

This manuscript presents an extensive and timely review of pretrained genomic language models (gLMs), providing a comprehensive comparison of machine learning architectures applied to genomic data. The work is well-structured, clearly written, and serves as a valuable reference for both computational and biological audiences. Overall, it represents a strong and meaningful contribution to the growing field of machine learning in genomics.

Strengths

1. Breadth and Depth of Comparative Analysis

The authors conducted an impressive and thorough comparative analysis, covering a wide range of architectures, data modalities, and pretraining strategies. The inclusion of both transformer-based and post-transformer architectures (such as HyenaDNA and state-space models) reflects a deep understanding of current trends in model development. This breadth adds real depth to the paper and positions it as a key reference for future gLM research.

2. Clarity and Accessibility

The manuscript makes concepts accessible to a non AI expert audience. The inclusion of glossaries, clear figures, and well-organized sections (from data modalities to benchmarking) demonstrates an effort to bridge computational and biological perspectives. This accessibility greatly enhances its reference value.

3. Structured Benchmarking and Evaluation

The review provides a strong overview of available benchmarks and evaluation strategies, including BEND, BEACON, and others. This section gives readers a comprehensive understanding of how different architectures perform across genomic tasks, highlighting the absence of a single "best" model, a valuable insight for practitioners.

Areas for Improvement

1. Discussion on Data Management and Preprocessing

While the manuscript provides coverage of tokenization and data modalities, it would benefit from a deeper discussion on data management and specially preprocessing. Specifically, it would be useful to address how factors such as curation of sequence, management of intergenic regions, repetitive sequences, and dataset composition influence model performance. A section examining the balance between generalization (cross-species or pan-genomic training) and specialization (domain- or task-specific models) would strengthen the practical relevance of the review.

2. Inclusion of Relevant Literature (AlphaGenome Preprint)

There is a notable omission of the AlphaGenome preprint from DeepMind, which is directly relevant to large-scale genomic foundation models and their pretraining strategies. Including this work would provide a more complete picture of the current landscape and ensure comprehensive coverage of recent advances. I strongly advise to integrate this work in the review.

3. Interpretability in Genomic Deep Learning

The manuscript could expand its discussion of interpretability. As gLMs become more complex, understanding their internal representations is crucial for biological trust and adoption. A summary of current interpretability efforts, including attention-based visualization, attribution methods, or latent space analysis, and their limitations would add significant value to the review.

4. Practical Considerations for Model Usability

While scalability and computational cost are addressed, the paper could include a concise section on practical deployment aspects. This might cover the computational resources required to train or fine-tune large gLMs, as well as the level of machine learning expertise needed to use them effectively. Discussing trade-offs between extensive architectures (e.g., Evo2) and smaller, domain-focused models (e.g., Caduceus or GENA-LM) would make the paper more useful for applied research settings.

5. Improving Narrative Flow and Emphasizing Key Takeaways

At times, the manuscript reads as an exhaustive list of models and methods, with limited guidance on which approaches perform best or are most promising for specific tasks. The authors could strengthen the narrative by explicitly highlighting the current best practices or "state-of-the-art" solutions for each major aspect, for example, which tokenization strategy achieves the best trade-off between compression and biological relevance, or which architectures scale most efficiently without loss of accuracy. Doing so would help readers navigate the dense comparisons and draw more actionable insights.

Summary

This manuscript is a strong and timely contribution to the field of deep learning in genomics. Its comprehensive overview of genomic language models, clarity of exposition, and structured benchmarking make it a valuable reference for computational biologists. Addressing the points above, particularly by enhancing discussions on data preprocessing, interpretability, usability, and by highlighting the most effective approaches in each evaluated aspect, would further enhance the manuscript tserve as an academic reference and a practical guide for real-world applications.

Reviewer #2:

Veiner and Supek provide a comprehensive overview of existing genomic language models (gLMs) and compare them across multiple aspects. I found the manuscript to be well-written, detailed, and mostly correct (see below). I have just a few comments:

1. Paragraph 499-505 is confusing. In line 499, the authors say "Encoder-only transformers mostly rely on the GPT", but GPT is a decoder-only model, so I assume that they wanted to write "Decoder-only transformers". However, in lines 504-505 they mention DNABERT2 and RiNALMo, which are encoder-only. If the reason for mentioning DNABERT2 and RiNALMo is their usage of FlashAttention, I suggest this be rephrased (or moved elsewhere), so that it is not assumed that they are also decoder-only.

Indeed, the reviewer is correct -- we meant to write decoder-only models. We have now split the mentioned poorly-phrased paragraph and moved it to more related parts of the same section, thus improving the narrative flow. We hope this prevents confusion. The text now reads:

PREVIOUS TEXT:

“On one hand, this constraint stems from the fact that these models are made position aware via fixed length positional encodings (the pairwise nature of the attention mechanism ignores positional information). Most other gLMs thus rely on Rotary Positional Embeddings (RoPE for short) (Su et al., 2024), as used in NT, AIDO.DNA, GENA-LM. RoPE allows the network to generalize to sequences longer than those it was pretrained on, and is the easiest off-the-shelf solution, as encodings are simply added to the token embeddings. Other positional encoding schemes such as Attention with Linear Biases (ALiBi) (preprint: Press et al., 2021) and relative positional encoding, used in T5 (Raffel et al., 2020). On the other hand, computing the full attention in every layer is costly, further constraining input size, which is why most transformer-based gLMs only process inputs of up to ~10kb (see Supplementary Table 1). Thus, recent advancements focused on reducing computational overhead via omitting the attention computation for some pairs of tokens, for example (Zaheer et al., 2020).

Encoder-only transformers mostly rely on the GPT (preprint: Radford et al., 2018) architecture, e.g. DNA-GPT, Llama (preprint: Touvron et al., 2023), e.g. METAGENE-1 (preprint: Liu et al., 2025b), or on Mistral (preprint: Jiang et al., 2023), e.g. GenomeOcean (preprint: Zhou et al., 2025). Computational optimization techniques were introduced over the years, the biggest of which is the introduction of flash attention (Dao et al., 2022), which is a hardware-aware algorithm for attention calculation in a more memory- and time-efficient manner, used in DNABERT2, and RiNALMo.

NEW TEXT FOLLOWS:

“Decoder-only transformers mostly rely on the GPT (preprint: Radford et al., 2018) architecture, e.g. DNA-GPT, Llama (preprint: Touvron et al., 2023), e.g. METAGENE-1 (preprint: Liu et al., 2025b), or on Mistral (preprint: Jiang et al., 2023), e.g. GenomeOcean (preprint: Zhou et al., 2025). However, the largest constraint of (encoder or decoder) transformer-based architectures is their limited context size.

On one hand, this constraint stems from the fact that BERT-based models are made position aware via fixed length positional encodings (the pairwise nature of the attention mechanism ignores positional information). Many gLMs thus rely on Rotary Positional Embeddings (RoPE for short) introduced in RoFormer (Su et al., 2024), as used in NT, AIDO.DNA, GENA-LM. RoPE allows the network to generalize to sequences longer than those it was pretrained on, and is the easiest off-the-shelf solution, as encodings are simply added to the token embeddings. Other positional encoding schemes such as Attention with Linear Biases (ALiBi) (preprint: Press et al., 2021) and relative positional encoding, used in T5 (Raffel et al., 2020). On the other hand, computing the full attention in every layer is costly,

further constraining input size, which is why most transformer-based gLMs only process inputs of up to ~10kb (see Dataset EV1). Thus, recent advancements focused on reducing computational overhead via omitting the attention computation for some pairs of tokens, for example (Zaheer et al., 2020), or leveraging hardware-aware algorithms for attention calculation, namely flash attention (Dao et al., 2022).”

2. While Tables 1 and 2 do a good job at presenting the comparison systematically, parts of this manuscript are text-heavy and somewhat difficult to read. The authors may consider creating one such (smaller) table for each section, e.g., Learning Objective, Architecture, etc. In my opinion, these tables would serve as summaries of each section and would help readers to better keep track of how different models operate

We thank the reviewer for the suggestion to reorganize the tables. Indeed we did that, albeit in a somewhat different organization than the reviewer suggested, which we thought does accomplish the goal of helping readers keep track: there is improved readability of the tables, while focusing on information relevant to the discussed aspects of gLMs.

Firstly, we reordered the tables to better follow the main text. Previously Table 1 “Independent benchmark datasets for evaluating genomic language models.” is now Table 3. Second previous Table 2 was split into two tables, now called “Table 1. Influential genomic language models across different data modalities.”, and “Table 2. Design choices and architectures of influential genomic language models.” All callouts in the text have been updated accordingly. Finally, we have also reformatted the text in the tables to decrease visual clutter.

3. gLMs are presented mostly as computational achievements and there is a lack of attention to experimental validation. For example, it is stated that RiNALMo outperforms other RNA foundation models, but the review doesn't address experimental confirmability, dataset biases, or whether predictions translate into functional assays.

In order to address reviewers comments 3. and 4. expanded on the “Criticisms of Genomic Language Models” section. Firstly, we renamed it to “Challenges and Future Developments”, containing three subsections, “Empirical Limitations of Current Genomic Language Models”, which contains most of the previous text of this section, the newly added “Experimental Confirmability and Evaluation Bias”, where we address reviewer comment 3, and “Developing Next Generation gLMs”. The new text under “Experimental Confirmability and Evaluation Bias” reads as follows:

“Many reported gains for gLMs remain confined to computational benchmarks, with few studies prospectively testing model-driven hypotheses in wet-lab assays. Positive examples demonstrate feasibility: Evo-generated sequences were shown to yield functional CRISPR-Cas molecular complexes as well as IS200/IS605 transposable systems (Nguyen et al., 2024), and UTR-LM prospectively validated predictions by testing a library of 211 engineered 5' UTRs with high predicted translation in reporter assays (Chu et al., 2024). A cost efficient alternative to this would be the in silico validation of model predictions via established state-of-the-art specialized models, for example using predicted structure via AlphaFold (Abramson et al., 2024). Evaluation efforts of gLM studies in practice rely on existing computational datasets and normally do not incorporate tasks that directly translate to outcomes of wet-lab experiments. Benchmarks using deep mutational scan, mutagenesis, and reporter data (Notin et al., 2023; Tang et al., 2025; (preprint: Robson & Ioannidis 2023) bring evaluations closer to DNA/RNA function than purely in silico metrics do, and should lead the field towards more applicable models.”

We also added sentence on dataset biases in a later paragraph:

“Moreover, commonly used pretraining and benchmark datasets exhibit nontrivial biases (e.g., taxonomic and assay over-representation, homologous sequences), which can inflate apparent performance and limit out-of-distribution generalization.”

4. The review lists several limitations of existing gLMs, (e.g., mutation insensitivity and long-range interaction). It would be interesting to see the analysis of why these failures persist and what conceptual innovations are required to address them.

We added the following text under the subsection “Developing Next Generation gLMs”, which addresses reviewer comments 4. and 6.

“These persistent limitations largely reflect a mismatch between current gLM assumptions and the structure of genomic information. Functional signals are sparse and highly context-dependent, yet reconstruction-style pretraining on unlabeled whole genomes treats all bases as equally informative, encouraging models to memorize local redundancy in repetitive DNA and overlook rare, cell-type-specific regulatory cues and single-nucleotide effects. Addressing this will likely require: (i) stronger biological inductive biases in pretraining (e.g., region-aware losses that prioritize functional bases and cis-regulatory elements; homology/evolution-informed objectives); (ii) mutational processes-aware tasks to improve single-base resolution modelling; (iii) hybrid or modular architectures that couple efficient long-range modeling with priors from regulatory genomics and structure; and (iv) multi-modal integration (e.g., genomic and histone modification signals) to supply missing domain context particularly with regard to local chromatin state.

The emergence of large-scale pangenome graphs offers an opportunity to expose models to population-level variation, structural variants, and haplotype context that have been largely absent from current training regimes. Unlocking these benefits will depend on advances in graph-aware representations, phasing-informed tokenization, and evaluation frameworks that account for redundancy and relatedness. We anticipate that such methods will be essential for enabling gLMs to learn efficiently from pangenome-scale diversity.”

5. Related to the previous point, the "Criticisms" section identifies issues such as catastrophic forgetting or weak mutation sensitivity, but does not weigh how critical they are relative to other fields (e.g., protein language models and NLP).

Catastrophic forgetting is also observed during finetuning textual LLMs, and it is still an active area of research on best practices to manage this. We have added a remark on this, and references on current articles (Luo et al., 2025; preprint Süalp & Rezaei, 2025).

Moreover, we added discussion on how pLMs are more sensitive to mutations than generally gLMs, as pLMs they relate mutations to structural (and so, presumably functional) changes on proteins. This suggests that in gLMs it should be beneficial to do the same using DNA-level functional outcomes, when possible, and we mention this in the text.

6. I would be interested to see if the authors could more explicitly give their opinion on how the field of gLMs will move forward. For example, will the future best-performing models also be mainly Transformers, or do you think it will hybrid architectures reach practical use? How will pangenome efforts reshape gLM training datasets.

As noted with reviewer comment 4., we have added a small section entitled “Developing Next Generation gLMs”, quoted above. Moreover at the end of the section “The Post-transformer Era” we added the following:

“Taken together, current results suggest that future best performing gLMs are unlikely to be pure Transformers, but will increasingly adopt hybrid designs that combine attention and/or local

convolution with state-space modules. We anticipate a division of labour: encoder-only Transformers will likely remain the default backbone for compact, task-focused gLMs, while large generalist and design-oriented models will converge on hybrid, potentially SSM-heavy stacks that scale quasi-linearly with sequence length.”

7. Typo in line 161: orgnaisms -> organisms

8. Line 166: GD -> GLM? If not, then where is the abbreviation GD defined?

Both typos have been corrected. We would like to thank the reviewer once more for the thoughtful comments.

Reviewer #3:

Review of "The DNA Dialect: A Comprehensive Guide to Pretrained Genomic Language Models"

Overview

This manuscript presents an extensive and timely review of pretrained genomic language models (gLMs), providing a comprehensive comparison of machine learning architectures applied to genomic data. The work is well-structured, clearly written, and serves as a valuable reference for both computational and biological audiences. Overall, it represents a strong and meaningful contribution to the growing field of machine learning in genomics.

Strengths

1. Breadth and Depth of Comparative Analysis

The authors conducted an impressive and thorough comparative analysis, covering a wide range of architectures, data modalities, and pretraining strategies. The inclusion of both transformer-based and post-transformer architectures (such as HyenaDNA and state-space models) reflects a deep understanding of current trends in model development. This breadth adds real depth to the paper and positions it as a key reference for future gLM research.

2. Clarity and Accessibility

The manuscript makes concepts accessible to a non AI expert audience. The inclusion of glossaries, clear figures, and well-organized sections (from data modalities to benchmarking) demonstrates an effort to bridge computational and biological perspectives. This accessibility greatly enhances its reference value.

3. Structured Benchmarking and Evaluation

The review provides a strong overview of available benchmarks and evaluation strategies, including BEND, BEACON, and others. This section gives readers a comprehensive understanding of how different architectures perform across genomic tasks, highlighting the absence of a single "best" model, a valuable insight for practitioners.

We thank the reviewer for summarizing the strengths of our article.

Areas for Improvement

1. Discussion on Data Management and Preprocessing

While the manuscript provides coverage of tokenization and data modalities, it would benefit from a deeper discussion on data management and specially preprocessing. Specifically, it would be useful to address how factors such as curation of sequence, management of intergenic regions, repetitive sequences, and dataset composition influence model performance. A section examining the balance between generalization (cross-species or pan-genomic training) and specialization (domain- or task-specific models) would strengthen the practical relevance of the review.

We have added a short section Data Curation and Generalization Trade-offs, which reads:

“Beyond modality, the curation of genomic data strongly influences gLM performance. A central challenge is the treatment of repetitive and intergenic regions: repetitive sequences, common in eukaryotes including plants (Zhai et al., 2025), are easy for language modelling yet contribute less functional signal. Some observations indicate that including intergenic data in the pretraining tasks can reduce downstream utility (preprint: Ellington et al., 2024; preprint: Wu et al., 2025). Mitigation strategies include focusing on functionally relevant or conserved regions (Benegas et al., 2025a), and down-weighting repetitive content (preprint: Ellington et al., 2024).

While models trained on diverse, multi-species datasets aim to leverage evolutionary information for broad generalization (Dalla-Torre et al., 2025), maximizing phylogenetic diversity is not universally optimal. Distinct genomic features may require specific evolutionary scopes: while deep timescales are helpful for coding and rare variants, functional non-coding and common variants are better captured at shallower timescales (preprint: Ye et al., 2025). Reflecting this need for specialization, region- or taxon-specific models can yield superior performance when sufficient pretraining data is available (Karollus et al., 2024, preprint: Benegas et al., 2025b, Fishman et al., 2025). Domain adaptation (see later) or continual pretraining may be used when such data is scarce or of low quality (preprint: Baghbanzadeh et al., 2025).”

2. Inclusion of Relevant Literature (AlphaGenome Preprint)

There is a notable omission of the AlphaGenome preprint from DeepMind, which is directly relevant to large-scale genomic foundation models and their pretraining strategies. Including this work would provide a more complete picture of the current landscape and ensure comprehensive coverage of recent advances. I strongly advise to integrate this work in the review.

Indeed, adding a mention of AlphaGenome was due, given its anticipated impact on the field. However since it is a supervised model, it falls somewhat outside the general scope of our review, which are unsupervised genomic models; we also explain this. See “Taxonomy of Genomic Language Models” section. We insert the relevant text below:

“In this section, we examine the pivotal design choices behind gLMs, including pretraining data, tokenizers, and architecture, and how they influence downstream performance. We restrict our analysis to self-supervised models trained on DNA or RNA sequences, explicitly excluding pLMs. While large sequence-to-function models like Enformer (Avsec et al., 2021), Borzoi (Linder et al., 2025), and BigRNA (preprint: Celaj et al., 2023) offer potential for domain-specific adaptation, they rely on supervised learning. This distinction also applies to the recently released AlphaGenome (preprint: Avsec et al., 2025). AlphaGenome unifies multimodal prediction to resolve diverse molecular phenotypes, such as splicing and chromatin states, at base-pair resolution from 1 Mb contexts. Unlike gLMs, which learn generalizable representations via self-supervision on raw DNA, these models are supervised directly on experimental functional tracks. Therefore, these architectures are not covered in this review.”

3. Interpretability in Genomic Deep Learning

The manuscript could expand its discussion of interpretability. As gLMs become more complex, understanding their internal representations is crucial for biological trust and adoption. A summary of current interpretability efforts, including attention-based visualization, attribution methods, or latent space analysis, and their limitations would add significant value to the review.

We added a full new section entitled “Interpretability Techniques”; to avoid redundancy, we removed some mention to techniques in other places. The added text is as follows:

“Several techniques have been proposed to investigate the internal workings of gLMs, commonly starting by an unsupervised clustering of sequence embeddings to assess whether the model’s learned representations align with known biological signals. These analyses frequently reveal clustering patterns that correspond to chromatin states, replication timing, and functional annotations (Nguyen et al., 2023; Dalla-Torre et al., 2025; Sanabria et al., 2024a; Chen et al., 2024).

Beyond static embeddings, the attention weights of transformer-based models can be visualized to highlight interactions between genomic regions, such as those between splicing donor and acceptor sites (Chen et al., 2024) or distinct regulatory elements (Dalla-Torre et al., 2025). We note, however, that transformers contain multiple attention matrices (one for each head across multiple layers) and so visualizing individual attention heads provides an incomplete picture. Researchers can also derive importance scores by masking input nucleotides and assigning weights based on the model’s predicted likelihood of the original base, as seen in GPN.

Perturbation-based approaches extend this idea by analyzing how changes in the input affect the output. Since gLMs output probability distributions, introducing point mutations at a specific position leads to shifts in nucleotide probabilities at other positions. Nucleotide dependency analysis utilizes this phenomenon to highlight important interactions between transcription factor binding sites and structural elements in RNA (Da Silva et al., 2025). A related approach, the categorical Jacobian (Zhang et al., 2024c), considers the sensitivity of the entire output logit matrix to mutations in the input sequence. While originally used for contact map prediction in protein language models (Zhang et al., 2024c), this method has recently been applied to analyze coevolutionary signals in gLMs (Nguyen et al., 2025).

These techniques are conceptually similar to “in silico mutagenesis” (ISM), often used to interrogate trained sequence-to-function models (Novakovsky et al., 2022). In ISM, each point mutation is given a score, based on the difference in predicted signals. This process can be sped up by hardware aware implementations of ISM, or by relying on the Taylor series approximation (Sasse et al., 2024). The resulting “attribution map” is well-suited for finding important sequence elements de novo, for example, by integrating with TF-MoDISco (Shrikumar et al., 2018). Recent methods such as SQUID (Seitz et al., 2024), and its successor SEAM (preprint: Seitz et al., 2025) offer an end-to-end pipeline for interpretability. These methods are also applicable to gLMs, just requires the construction of an attribution map based on its outputs.

Finally, the most recent trend involves adapting mechanistic interpretability techniques from NLP (Cunningham et al., 2023) by using sparse autoencoders to decompose dense model representations. This method has been applied to Evo 2 to isolate specific directions in the latent space that correspond to high-resolution biological features, such as intron-exon boundaries and transcription factor motifs (preprint: Brixi et al., 2025).”

4. Practical Considerations for Model Usability

While scalability and computational cost are addressed, the paper could include a concise section on practical deployment aspects. This might cover the computational resources required to train or fine-tune large gLMs, as well as the level of machine learning expertise needed to use them effectively. Discussing trade-offs between extensive architectures (e.g., Evo2) and smaller, domain-focused models (e.g., Caduceus or GENA-LM) would make the paper more useful for applied research settings.

We have added a section on “Practical considerations”, the text reads:

”The barrier to entry for deploying genomic language models has been significantly lowered by the standardization of model repositories. Most current architectures are readily available via platforms like Hugging Face, allowing researchers to perform inference and training using standardized pipelines, and significantly lowering the barrier to entry. The computational burden can be further managed by previously mentioned PEFT strategies. This approach enables the fine-tuning of substantial models on consumer-grade hardware rather than requiring datacenter-scale infrastructure.

Selecting the appropriate gLM requires balancing the scope of the biological inquiry against computational resources available. Extensive generalist architectures, such as Evo 2, capture diverse genomic patterns, but they impose high resource demands. In contrast, domain-focused models like Species-LM or Orthrus prioritize efficiency, but have architectural priors tailored to specific biological properties. These specialized models often yield competitive performance on targeted downstream tasks with significantly greater computational efficiency, making them highly suitable for applied research settings.”

5. Improving Narrative Flow and Emphasizing Key Takeaways

At times, the manuscript reads as an exhaustive list of models and methods, with limited guidance on which approaches perform best or are most promising for specific tasks. The authors could strengthen the narrative by explicitly highlighting the current best practices or "state-of-the-art" solutions for each major aspect, for example, which tokenization strategy achieves the best trade-off between compression and biological relevance, or which architectures scale most efficiently without loss of accuracy. Doing so would help readers navigate the dense comparisons and draw more actionable insights.

We agree with the reviewer and to address this, we have added an additional Box, entitled “Practitioner’s Short Guide to Selecting the Right gLM”. The text is as follows:

“Variant effect prediction. Prioritize models trained with single-nucleotide tokens, and with evolutionarily-informed objectives (e.g. GPN family, Orthrus, NT-multispecies). Avoid BPE-based models where the variant might be merged into a larger token.

Gene/genomic element annotation. DNABERT-2 and GROVER, based on BPE tokenization, remain robust generalist choices for identifying promoters and functional elements in short-to-medium contexts.

Long-range Tasks (approximately >10kb). Select SSM-based (HyenaDNA, Caduceus) or hybrid (e.g. Evo 2) architectures. Transformer-based models may become computationally prohibitive and show diminishing returns at this scale.

Limited compute and/or large datasets. Use domain-specialized models (e.g., Species-LM for fungi, Orthrus for RNA) which often outperform larger generalist models on their specific tasks while requiring a fraction of the inference cost.”

We have also streamlined the text, primarily in the early parts within “Taxonomy of Genomic Language Models” at several places to improve the narrative flow.

Summary

This manuscript is a strong and timely contribution to the field of deep learning in genomics. Its comprehensive overview of genomic language models, clarity of exposition, and structured benchmarking make it a valuable reference for computational biologists. Addressing the points above, particularly by enhancing discussions on data preprocessing, interpretability, usability, and by highlighting the most effective approaches in each evaluated aspect, would further enhance the manuscript to serve as an academic reference and a practical guide for real-world applications.

References

We added the following references relevant to the discussion of reviewer comments:

“Abramson J, Adler J, Dunger J, Evans R, Green T, Pritzel A, Ronneberger O, Willmore L, Ballard AJ, Bambrick J, *et al* (2024) Accurate structure prediction of biomolecular interactions with AlphaFold 3. *Nature* 630: 493–500

Avsec Ž, Latysheva N, Cheng J, Novati G, Taylor KR, Ward T, Bycroft C, Nicolaisen L, Arvaniti E, Pan J, *et al* AlphaGenome: advancing regulatory variant effect prediction with a unified DNA sequence model.

Cunningham H, Ewart A, Riggs L, Huben R & Sharkey L (2023) Sparse Autoencoders Find Highly Interpretable Features in Language Models. doi:10.48550/arXiv.2309.08600 [PREPRINT]

Luo Y, Yang Z, Meng F, Li Y, Zhou J & Zhang Y (2025b) An Empirical Study of Catastrophic Forgetting in Large Language Models During Continual Fine-tuning. doi:10.48550/arXiv.2308.08747 [PREPRINT]

Novakovsky G, Dexter N, Libbrecht MW, Wasserman WW & Mostafavi S (2023) Obtaining genetics insights from deep learning via explainable artificial intelligence. *Nat Rev Genet* 24: 125–137

Sasse A, Chikina M & Mostafavi S (2024) Quick and effective approximation of in silico saturation mutagenesis experiments with first-order Taylor expansion. *iScience* 27: 110807

Seitz EE, McCandlish DM, Kinney JB & Koo PK Uncovering the Mechanistic Landscape of Regulatory DNA with Deep Learning. doi:10.1101/2025.10.07.681052 [PREPRINT]

Shrikumar A, Tian K, Avsec Ž, Shcherbina A, Banerjee A, Sharmin M, Nair S & Kundaje A Technical Note on Transcription Factor Motif Discovery from Importance Scores (TF-MoDISco) version 0.5.6.5. doi: arXiv:1811.00416 [PREPRINT]

Süalp E & Rezaei M (2025) Mitigating Catastrophic Forgetting in Continual Learning through Model Growth. doi:10.48550/arXiv.2509.01213 [PREPRINT]

Ye C, Benegas G, Albors C, Li JC, Prillo S, Fields PD, Clarke B & Song YS Predicting functional constraints across evolutionary timescales with phylogeny-informed genomic language models. doi: 10.1101/2025.09.21.677619 [PREPRINT]”

We also updated the following ones from preprint to final version:

“Karollus A, Hingerl J, Galindez GStaT, Wagner N, Hernandez-Alias X, Incarnato D & Gagneur J (2025) Nucleotide dependency analysis of genomic language models detects functional elements. *Nat Genet* 57: 2589–2602

De Almeida BP, Dalla-Torre H, Richard G, Blum C, Hexemer L, Gélard M, Mendoza-Revilla J, Tang Z, Marin FI, Emms DM, *et al* (2025) Annotating the genome at single-nucleotide resolution with DNA foundation models. *Nat Methods* 22: 2301–2315

Yin W, Zhang Z, Zhang S, He L, Zhang R, Jiang R, Liu G, Wang J, Zhang X, Qin T, *et al* (2025) ERNIE-RNA: an RNA language model with structure-enhanced representations. *Nat Commun* 16: 10076”

2nd Dec 2025

Manuscript number: MSB-2025-13304R

Title: The DNA Dialect: A Comprehensive Guide to Pretrained Genomic Language Models

Dear Fran,

Thank you again for submitting your revised manuscript. I am pleased to inform you that your paper has been accepted for publication.

I have forwarded your figures to our designer, who will contact you directly regarding the redrawn images once they are ready.

After you approve the redrawn images, your manuscript will be exported to our production team to begin publication processing. It will undergo copy editing, and you will receive page proofs prior to publication.

Please note that you will be contacted by Springer Nature Author Services to complete licensing and payment information. When prompted by the Author Services system, please use the following token for payment: Token NDIXNZK1MDA0.

Should you be planning a Press Release on your article, please get in contact with embo_production@springernature.com after the manuscript is exported, in order to coordinate publication and release dates.

Thank you for your valuable contribution to Molecular Systems Biology!

Kind regards,
Jingyi

Jingyi Hou, PhD
Senior Editor
Molecular Systems Biology
